# Role of cyclooxygenase-2-mediated prostaglandin E2-prostaglandin E receptor 4 signaling in cardiac reprogramming

Naoto Muraoka[1], Kaori Nara[1], Fumiya Tamura[1], Hidenori Kojima[1], Hiroyuki Yamakawa[1], Taketaro Sadahiro[2], Kazutaka Miyamoto[1], Mari Isomi[2], Sho Haginiwa[1], Hidenori Tani[1], Shota Kurotsu[1], Rina Osakabe[1], Satoru Torii[3], Shigeomi Shimizu[3], Hideyuki Okano[4], Yukihiko Sugimoto [5], Keiichi Fukuda [1] & Masaki Ieda[2]

Direct cardiac reprogramming from fibroblasts can be a promising approach for disease modeling, drug screening, and cardiac regeneration in pediatric and adult patients. However, postnatal and adult fibroblasts are less efficient for reprogramming compared with embryonic fibroblasts, and barriers to cardiac reprogramming associated with aging remain undetermined. In this study, we screened 8400 chemical compounds and found that diclofenac sodium (diclofenac), a non-steroidal anti-inflammatory drug, greatly enhanced cardiac reprogramming in combination with *Gata4*, *Mef2c*, and *Tbx5* (GMT) or GMT plus *Hand2*. Intriguingly, diclofenac promoted cardiac reprogramming in mouse postnatal and adult tail-tip fibroblasts (TTFs), but not in mouse embryonic fibroblasts (MEFs). Mechanistically, diclofenac enhanced cardiac reprogramming by inhibiting cyclooxygenase-2, prostaglandin E2/prostaglandin E receptor 4, cyclic AMP/protein kinase A, and interleukin 1β signaling and by silencing inflammatory and fibroblast programs, which were activated in postnatal and adult TTFs. Thus, anti-inflammation represents a new target for cardiac reprogramming associated with aging.

[1] Department of Cardiology, Keio University School of Medicine, 35 Shinanomachi, Shinjuku-ku, Tokyo 160-8582, Japan. [2] Department of Cardiology, Faculty of Medicine, University of Tsukuba, 1-1-1 Tennoudai, Tsukuba City, Ibaraki 305-8575, Japan. [3] Department of Pathological Cell Biology, Medical Research Institute, Tokyo Medical and Dental University (TMDU), 1-5-45 Yushima, Bunkyo-ku, Tokyo 113-8510, Japan. [4] Department of Physiology, Keio University School of Medicine, 35 Shinanomachi, Shinjuku-ku, Tokyo 160-8582, Japan. [5] Department of Pharmaceutical Biochemistry, Graduate School of Pharmaceutical Sciences, Kumamoto University, 5-1 Oehoncho, Chuo-ku, Kumamoto-shi, Kumamoto 862-0973, Japan. Correspondence and requests for materials should be addressed to M.I. (email: mieda@md.tsukuba.ac.jp)

Heart disease is a leading cause of mortality in children and adults, and new therapeutic options are in high demand. Direct cardiac reprogramming converts fibroblasts into cardiomyocyte-like cells (iCMs), without passing through a stem cell state, by the overexpression of cardiac transcription factors including *Gata4*, *Mef2c*, and *Tbx5* (GMT) or GMT plus *Hand2* (GHMT)[1,2]. In vivo cardiac reprogramming by direct injection of GMT or GHMT into infarct mouse hearts converted resident cardiac fibroblasts into iCMs, improved cardiac function, and reduced fibrosis after myocardial infarction (MI)[2–5]. Zhou et al.[6] recently reported that comparative gene expression analyses showed iCMs induced in vitro exhibited more adult cardiomyocyte-like features, such as fatty acid oxidation and cell-cycle exit, than exhibited by induced pluripotent stem cell (iPSC)-derived CMs. Thus, direct cardiac reprogramming has potential for disease modeling, drug screening, and cardiac repair, if the iCMs can be efficiently generated from fibroblasts[7].

We and others have mainly taken a candidate approach to identify the factors that enhance cardiac reprogramming. Recent advances in this field have shown that modifications of transcription factors, miRNAs, epigenetic factors, defined culture conditions, and small molecules (including TGFβ Wnt inhibitors), could promote cardiac reprogramming[8–15]. Although silencing the fibroblast (original cell type) program is a prerequisite for cardiac reprogramming, the molecular mechanisms underlying this process remain poorly understood. Moreover, improvements in reprogramming efficiency were shown mainly in mouse embryonic fibroblasts (MEFs), and cardiac reprogramming from more differentiated fibroblasts, such as mouse postnatal and adult tail-tip fibroblasts (TTFs), remained inefficient[13,16]. For clinical relevance, it is desirable to generate iCMs efficiently from postnatal and adult fibroblasts; however, the barriers to cardiac reprogramming associated with aging remain undefined[7,17].

In this study, we developed a high-content, high-throughput screening system, using a chemical library of 8400 compounds, to identify small molecules that enhance cardiac reprogramming in mouse postnatal and adult TTFs. Small molecules would be less expensive, more easily controlled, and possibly more efficient than growth factors and cytokines, leading to efficient and reproducible cardiac reprogramming. In this study, we found diclofenac sodium (diclofenac) greatly enhanced cardiac reprogramming in postnatal and adult TTFs, but not in MEFs, in combination with GMT or GHMT. Diclofenac enhanced cardiac reprogramming via the inhibition of cyclooxygenase-2 (COX-2)/ prostaglandin E2 (PGE2)/PGE receptor 4 (EP4)/interleukin 1β (IL-1β)/interleukin 1 receptor type 1 (IL-1R1) signaling and subsequent suppression of inflammatory and fibroblast gene programs, which were activated in postnatal and adult fibroblasts.

## Results

### Diclofenac promoted cardiac reprogramming in postnatal TTFs. 
We previously identified cardiac reprogramming factors, *Gata4*, *Mef2c*, and *Tbx5*, by analyzing cardiac reporter and protein expression in mouse fibroblasts with fluorescence-activated cell sorting (FACS) analyses[1]. However, this method is labor-intensive and may not be suitable to screen a large set of chemical compounds. To screen small molecules that promote cardiac reprogramming from chemical libraries, we developed a high-content, high-throughput screening system, using an IN Cell Analyzer 6000. We realized that fibroblasts from alpha myosin heavy chain (αMHC)-GFP transgenic mice exhibited auto-fluorescence in this system, hindering accurate detection of cardiac reprogramming with GFP signals. Consequently, we

generated αMHC-Cre/R26-tdTomato mice by crossing αMHC-Cre mice with R26R-tdTomato reporter mice, in which only cardiomyocytes expressed tdTomato. We used αMHC-Cre/R26-tdTomato mouse postnatal TTFs, which were not contaminated with cardiomyocytes and did not express auto-fluorescence, to identify chemicals that promote cardiac reprogramming in differentiated fibroblasts. For chemical screening, we transduced GMT retroviruses into αMHC-Cre/R26-tdTomato TTFs in multiple 96-well plates and added 8400 small molecules at a final concentration of 10 μM in each well. After 1 week, cell nuclei were stained with Hoechst 33342, and cardiac reprogramming efficiency was automatically scored by the IN Cell Analyzer 6000 as the percentage of αMHC-Cre/R26-tdTomato⁺ nuclei out of the total number of nuclei (Fig. 1a). The first screening identified 37 hit compounds that increased cardiac reprogramming efficiency more than the mean + 2 SD of the control (Fig. 1b). The second round of screening narrowed the 37 compounds down to 4, including QA-6161, STK081095, STK833231, and STK825882 (Fig. 1b, c, Supplementary Table 2). Next, to confirm the effects of these 4 compounds on cardiac reprogramming under different conditions, we transduced GHMT into αMHC-GFP TTFs, treated the cells with the four compounds, and after 1 week analyzed αMHC-GFP cardiac reporter and endogenous cardiac troponin T (cTnT) protein expression with FACS. FACS analyses showed that all four chemicals increased the generation of αMHC-GFP⁺ and cTnT⁺ cells from postnatal TTFs (Fig. 1d, e). Intriguingly, the most powerful compound, QA-6161, was proven to be diclofenac (Voltaren), an FDA-approved non-steroidal anti-inflammatory drug (NSAID), commonly used in humans for the treatment of inflammation and rheumatic diseases (Fig. 1f), while the functions of the other compounds remain largely unknown. Addition of each compound individually to diclofenac did not show additive effects on cardiac reprogramming, suggesting these three compounds regulated the same signaling pathways as diclofenac (Supplementary Fig. 1a, b). Therefore, we focused on diclofenac in subsequent studies to determine its effects on cardiac reprogramming

### Diclofenac functioned at the early stage of reprogramming. 
FACS analyses showed that 10 μM of diclofenac with GHMT increased cardiac reprogramming efficiency in postnatal TTFs about three- to fourfold, reaching a comparable level with that in MEFs (Figs. 2a, b and 3c, d). We next analyzed the dose dependency of diclofenac-mediated cardiac reprogramming in αMHC-GFP postnatal TTFs with FACS. Titration of diclofenac from 1 nM to 100 μM improved cardiac reprogramming in a dose-dependent manner without cytotoxicity (Fig. 2a, b). Cardiac induction was efficient even at the low concentration (1 μM), demonstrating the strong biological activity of this compound. To determine the period during which diclofenac promoted cardiac reprogramming, we cultured the cells with diclofenac in several patterns and analyzed cardiac induction after 7 days with FACS. Diclofenac treatment on days 0–4 was similarly effective as that on days 0–7, while treatment after day 4 (days 4–7) did not improve cardiac reprogramming. This suggests that diclofenac is critical for iCM generation during the early stages of cardiac reprogramming, in which the fibroblast program is silenced (Fig. 2c). Immunostaining for cardiac markers showed that diclofenac increased the generation of αMHC-GFP⁺ and α-actinin⁺ cells in the GHMT-transduced TTFs after 4 weeks, and the GHMT/diclofenac-induced iCMs had well-defined sarcomeric structures (Fig. 2d, e). A quantitative reverse transcription-polymerase chain reaction (qRT-PCR) showed that a panel of cardiac genes related to different functions, including sarcomeric structures (*Actc1*, *Myh6*, *Tnnc1*), hormones (*Nppa*), and ion

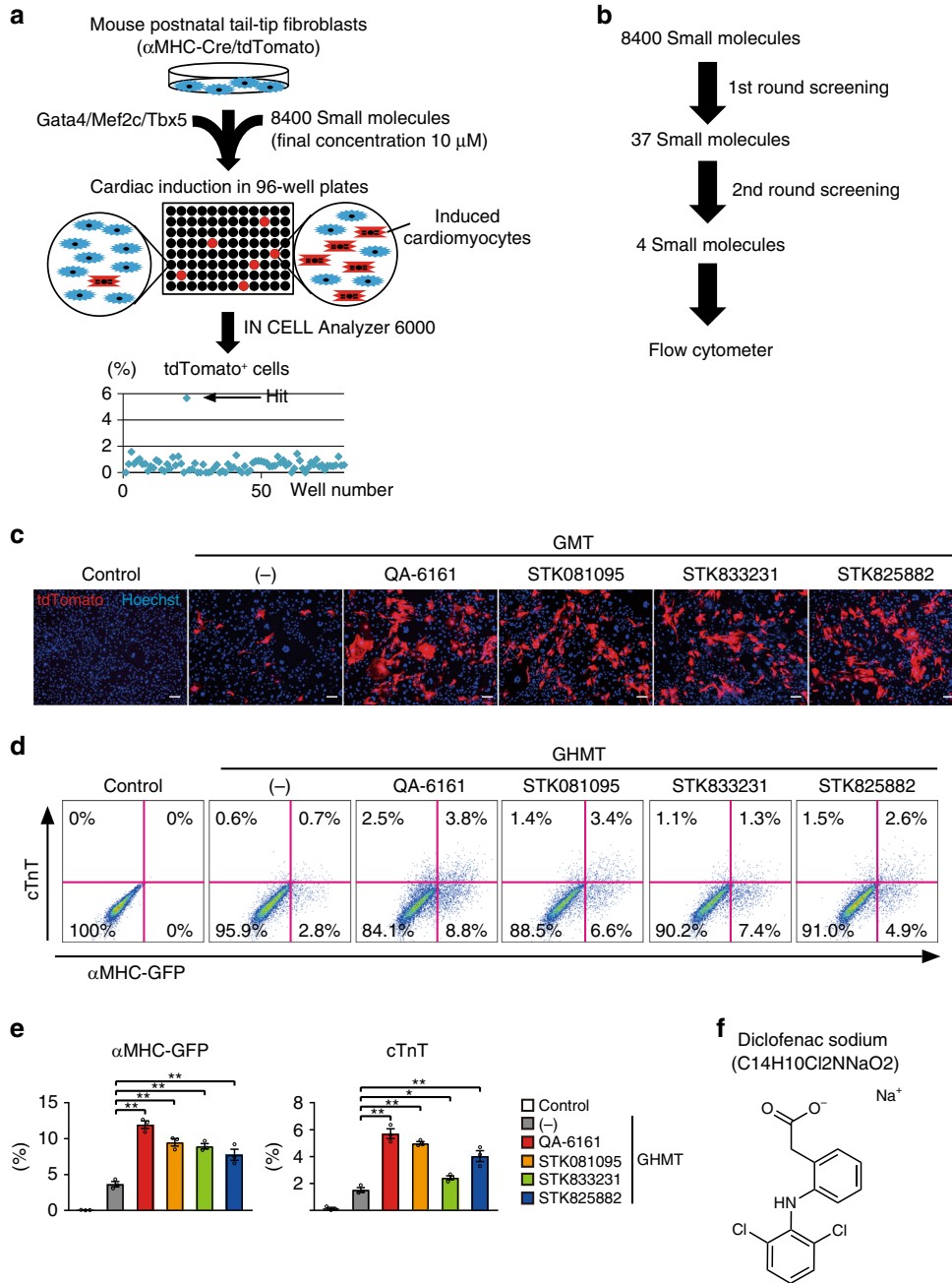

**Fig. 1** High-content screening identifies small molecules for cardiac reprogramming. **a**, **b** Schematic representation of the high-content, high-throughput screening system designed to identify small molecules for cardiac reprogramming in αMHC-Cre/R26-tdTomato TTFs in the presence of GMT. After the second screening, FACS was performed (**b**). **c** Four compounds increased generation of αMHC-Cre/R26-tdTomato+ cells from postnatal TTFs after 1 week. See also Supplementary Table 2. **d**, **e** FACS analyses for αMHC-GFP and cTnT expression in GHMT-transduced TTFs treated with indicated compounds for 1 week. Quantitative data are shown in **e**; $n = 3$ biologically independent experiments. See also Supplementary Figure 1. **f** Chemical structure of diclofenac (QA-6161). One-way ANOVA with Dunnett's post hoc test was performed for **e**; all data are presented as mean ±SEM. *$P < 0.05$, **$P < 0.01$ vs. the relevant control. Scale bars represent 100 μm

channels (*Kcnd2*, *Pln*, *Scn5a*, *Slc8a1*), were strongly upregulated in the reprogrammed TTFs with diclofenac (Fig. 2f).

We next analyzed the functional property characteristics of cardiomyocytes in the GHMT/diclofenac-induced iCMs. To detect the cells that exhibited spontaneous calcium transients, we analyzed intracellular calcium concentrations by Rhod-3 imaging (Fig. 2g, Supplementary Movie 1). We found that the combination of GHMT and diclofenac after 6 weeks increased the generation of spontaneous calcium transient+ cells about two- to

threefold when compared to GHMT alone (Fig. 2h). The number of spontaneously beating iCMs in GHMT-TTFs after 6 weeks also increased about fourfold with the addition of diclofenac (Fig. 2i, Supplementary Movie 2). Therefore, diclofenac promoted the efficiency and quality of cardiac reprogramming in mouse postnatal TTFs.

**Diclofenac enhanced reprogramming via suppression of COX-2.** We next investigated the effect of diclofenac on cardiac

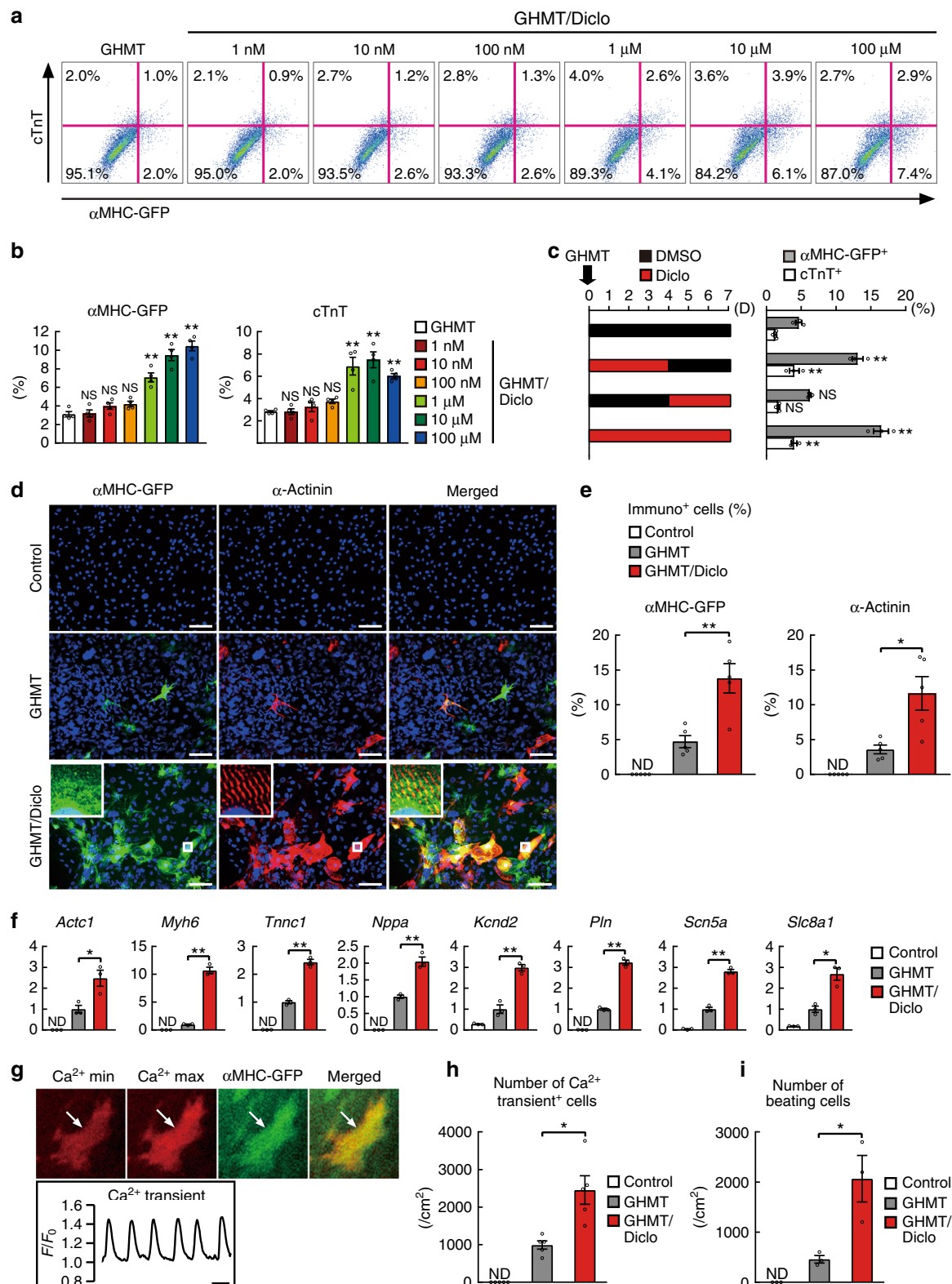

reprogramming in adult TTFs. FACS analyses showed that cardiac reprogramming with GHMT was less efficient in adult TTFs than in postnatal TTFs, consistent with previous reports (Figs. 2a, b and 3a, b)[13,16]. However, diclofenac treatment significantly increased the induction of αMHC-GFP+ and cTnT+ cells (by about threefold) to a level comparable to that in postnatal TTFs

(Figs. 2a, b and 3a, b). Moreover, addition of miR-133, a muscle miRNA known to promote cardiac reprogramming, to GHMT/ diclofenac further enhanced cardiac induction in adult TTFs to a similar extent as that in MEFs and diclofenac-treated postnatal TTFs (Figs. 2a, b and 3c, d, Supplementary Fig. 2a, b). We next analyzed the effect of diclofenac on cardiac reprogramming in

**Fig. 2** Diclofenac promotes the efficiency and quality of cardiac reprogramming in TTFs. **a, b** Dose dependency of Diclofenac-mediated cardiac reprogramming in postnatal αMHC-GFP TTFs in combination with GHMT. Quantitative data are shown in **b**; $n = 4$ biologically independent experiments. **c** GHMT-transduced TTFs were cultured with (red bars, left) or without (black bars) diclofenac as indicated for 1 week. FACS analyses for αMHC-GFP and cTnT expression in three independent triplicate experiments are shown (right). D indicates days. **d, e** Immunocytochemistry for αMHC-GFP, α-actinin, and DAPI. GHMT/diclofenac induced more αMHC-GFP and α-actinin expression than GHMT alone in TTFs after 4 weeks. High-magnification views in insets show the sarcomeric organization. Quantitative data are shown in **e**; $n = 5$ biologically independent experiments. **f** QRT-PCR for cardiac gene expression in the control-, GHMT-, and GHMT/diclofenac-treated TTFs after 1 week; $n = 3$ biologically independent experiments. **g** Spontaneous $Ca^{2+}$ oscillations in the GHMT/diclofenac-induced iCMs at 6 weeks after transduction. Maximum and minimum concentrations of $Ca^{2+}$ signals are shown in the upper panels, and the Rhod-3 intensity trace is shown in the lower panel. See also Supplementary Movie 1. **h** Quantitative data of the number of $Ca^{2+}$ oscillation+ cells after 6 weeks are shown; $n = 5$ biologically independent experiments. **i** Quantitative data of the number of spontaneously beating cells in control-, GHMT-, and GHMT/diclofenac-treated TTFs after 6 weeks are shown; $n = 3$ biologically independent experiments. See also Supplementary Movie 2. Student's $t$-test was performed for **e, f, g, i**, one-way ANOVA with Dunnett's post hoc test was performed for **b, c**; all data are presented as mean ±SEM. *$P < 0.05$, **$P < 0.01$ vs. the relevant control. ND, not detected. Diclo, diclofenac. Scale bars represent 100 μm (**d**) and 1 s (**g**)

MEFs. In contrast to the positive effects in postnatal and adult TTFs, diclofenac did not substantially promote cardiac reprogramming in GHMT-transduced MEFs (Fig. 3c, d). These results suggest that diclofenac promotes cardiac reprogramming selectively in mouse postnatal and adult TTFs, but not in MEFs.

Diclofenac and other NSAIDs inhibit the activity of COX enzymes, and thereby inhibit the biosynthesis of prostanoids[18]. Two COX isoforms are known: COX-1, which is constitutively expressed, and COX-2, which is inducible by inflammatory growth factors and cytokines[19]. Diclofenac inhibits COX-2 activity more strongly than COX-1[20]. Next, we analyzed each COX gene expression in MEFs, postnatal and adult TTFs, and in postnatal mouse hearts. QRT-PCR analyses showed that *Ptgs1* (COX-1) expression was two- to threefold higher in postnatal and adult TTFs than in MEFs and heart samples. Notably, *Ptgs2* (COX-2) was strongly expressed in postnatal and adult TTFs compared to MEFs in an age-dependent manner and was barely detected in postnatal heart samples (Fig. 3e). Consistently, we found that multiple inflammatory and fibroblast-related genes, including prostaglandin E receptor 4 (*Ptger4*), MCP-1 (*Ccl2*), MCP-2 (*Ccl8*), IL-1R1 (*Il1r1*), *Postn*, and *Thy1*, were more strongly expressed in postnatal and adult TTFs than in MEFs (Fig. 3e). To investigate which COX isoform was critical for diclofenac-mediated cardiac reprogramming, we treated the GHMT-transduced postnatal TTFs with a COX-1 (SC-560) or COX-2 (NS-398) specific inhibitor. FACS analyses revealed that, like diclofenac, NS-398 (but not SC-560) greatly increased the generation of αMHC-GFP+ and cTnT+ cells in the GHMT-transduced TTFs (Fig. 3f, g). Of the other NSAIDs tested, celecoxib (a COX-2 selective inhibitor) but not acetylsalicylic acid (aspirin, a COX-1 dominant inhibitor), also improved cardiac reprogramming (Fig. 3h–k). These results suggest that diclofenac promotes cardiac reprogramming in TTFs mainly through the suppression of COX-2, which was highly expressed in postnatal and adult TTFs.

**Diclofenac promoted reprogramming by suppressing PGE2/EP4**. COX enzymes synthesize prostanoids, which consist of PGD2, PGE2, PGF2α, PGI2, and thromboxane A2 (TXA2)[18]. To determine the molecular mechanism underlying diclofenac-mediated cardiac reprogramming, each prostanoid was added to the GHMT/diclofenac-induced TTFs. We found that the addition of PGE2 strongly suppressed the generation of GHMT/diclofenac-induced iCMs compared to other prostanoids, suggesting suppression of PGE2 synthesis was critical for diclofenac-mediated cardiac reprogramming (Fig. 4a, b). PGE2 binds to four kinds of G-protein-coupled receptors, PGE receptors 1–4 (EP1-4, *Ptger1-4*)[18,19]. Although all four PGE receptors were expressed in postnatal TTFs, *Ptger4* was most abundantly

expressed in TTFs (Supplementary Fig. 3a). To determine which PGE receptors were involved in cardiac reprogramming, we cultured GHMT-transduced postnatal TTFs with specific antagonists for EP1 (ONO-8713), EP2 (TG4-155), EP3 (ONO-AE5-599), or EP4 (ONO-AE3-208). FACS analyses revealed that the EP4 antagonist most strongly induced αMHC-GFP+ and cTnT+ cells, while EP3 antagonist treatment showed a mild effect. Addition of EP3 antagonist to EP4 antagonist did not further promote cardiac reprogramming, suggesting that EP3 shared the same downstream signaling pathways as EP4 (Figs. 4c–e and 5i, Supplementary Fig. 3b). We next suppressed EP4 (*Ptger4*) expression with siRNA in GHMT-transduced TTFs (Supplementary Fig. 3c). FACS analyses showed that knockdown of *Ptger4* also increased cardiac reprogramming from postnatal TTFs, recapitulating the effect of diclofenac (Supplementary Fig. 3d, e). In contrast, similar to PGE2 treatment, the addition of the EP4 selective agonist (ONO-AE1-329) completely blocked diclofenac-mediated cardiac reprogramming, suggesting that EP4 is a major receptor involved in diclofenac-induced cardiac reprogramming (Supplementary Fig. 3f, g). Next, to confirm the role of EP4 in cardiac reprogramming, we used EP4-knockout mice (*Ptger4*$^{-/-}$). Cardiac reprogramming was greatly increased in *Ptger4*$^{-/-}$ TTFs to a similar extent as that in EP4 antagonist/diclofenac-treated wild-type TTFs. Addition of EP4 antagonist, diclofenac, or PGE2 did not change cardiac induction in *Ptger4*$^{-/-}$ TTFs, suggesting that EP4 is critical for diclofenac/PGE2-mediated cardiac reprogramming (Fig. 4f, g). Consistent with the cell type-specific effects of diclofenac on cardiac reprogramming, *Ptger4* was more highly expressed in postnatal and adult TTFs than in MEFs (Fig. 3e). These results suggest that diclofenac promotes cardiac reprogramming in TTFs mainly through the inhibition of PGE2/EP4 signaling.

**cAMP/PKA activation pathway blocked cardiac reprogramming**. After 4 weeks of treatment, we analyzed the effects of the PGE2/EP4 pathway on diclofenac-induced cardiac reprogramming using immunohistochemistry in postnatal and adult TTFs. Immunostaining for the cardiac markers αMHC-GFP, cTnT, and α-actinin showed that PGE2 addition strongly suppressed GHMT/diclofenac-induced cardiac reprogramming in postnatal and adult TTFs (Fig. 5a, b, Supplementary Fig. 4a, b). After 4 weeks of treatment, the EP4 antagonist (ONO-AE3-208) strongly enhanced iCM generation, whereas the EP4 specific agonist (ONO-AE1-329) or PGE2 suppressed diclofenac-mediated cardiac reprogramming, consistent with the FACS data (Fig. 5c, d, Supplementary Fig. 4a, b). The qRT-PCR analyses also revealed that a panel of cardiac genes, including *Tnnc1*, *Nppa*, *Gja1*, *Kcnd2*, *Pln*, and *Scn5a*, were upregulated with the addition of the EP4 antagonist and downregulated with the

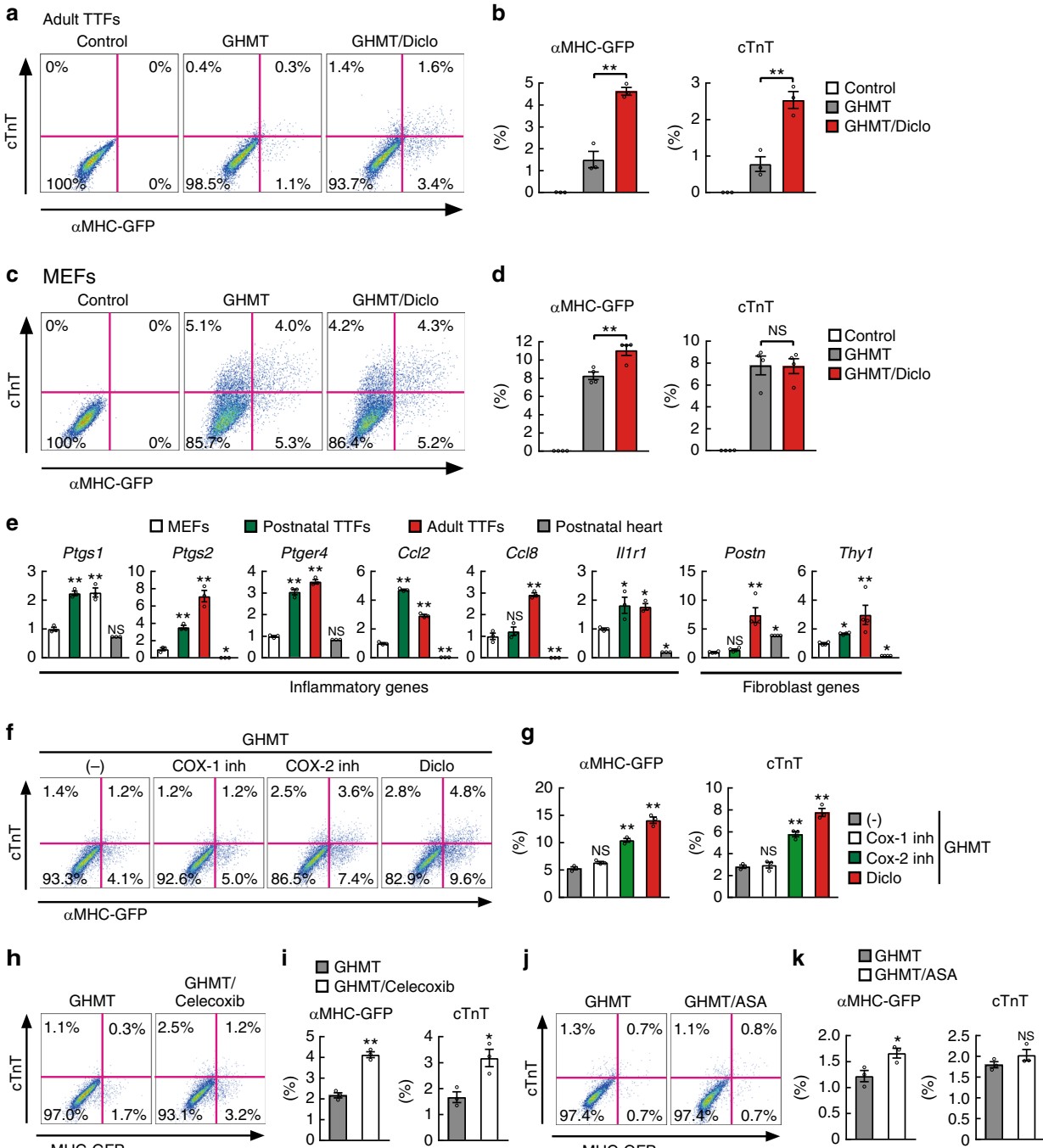

**Fig. 3** Diclofenac promotes cardiac reprogramming in adult TTFs but not in MEFs. **a**, **b** FACS analyses for αMHC-GFP and cTnT expression in adult TTFs transduced with GHMT and cultured with or without diclofenac for 1 week. Quantitative data are shown in **b**; n = 3 biologically independent experiments. See also Supplementary Figure 2. **c**, **d** FACS analyses for αMHC-GFP and cTnT expression in MEFs transduced with GHMT and cultured with or without diclofenac for 1 week. Quantitative data are shown in **d**; n = 3 biologically independent experiments. **e** Relative mRNA expression levels were determined in MEFs, postnatal and adult TTFs, and postnatal heats by qRT-PCR; n = 3 biologically independent experiments. **f**, **g** FACS analyses for αMHC-GFP and cTnT expression in postnatal TTFs transduced with GHMT and treated with indicated reagents for 1 week; inh indicates inhibitor. Quantitative data are shown in **g**; n = 3 biologically independent experiments. **h**–**k** FACS analyses for αMHC-GFP and cTnT expression in postnatal TTFs transduced with GHMT and treated with celecoxib (**h**) or acetylsalicylic acid (ASA, **j**) for 1 week. Quantitative data are shown in **i** and **k**, respectively; n = 3 biologically independent experiments. Student's t-test was performed for **b**, **d**, **i**, **k**, one-way ANOVA with Dunnett's post hoc test was performed for **e**, **g**; all data are presented as mean ± SEM. *P < 0.05, **P < 0.01 vs. the relevant control. NS, not significant. Diclo, diclofenac

addition of the EP4 agonist or PGE2 in both TTFs (Fig. 5e, Supplementary Fig. 4c), suggesting that diclofenac promotes cardiac reprogramming in postnatal and adult TTFs through the inhibition of PGE2/EP4 signaling.

PGE2/EP4 signaling activates the cAMP/PKA pathway through Gs, however, the role of the cAMP/PKA pathway in cardiac reprogramming remains undetermined[18]. Forskolin activates adenylyl cyclase and increases the intracellular level of cAMP,

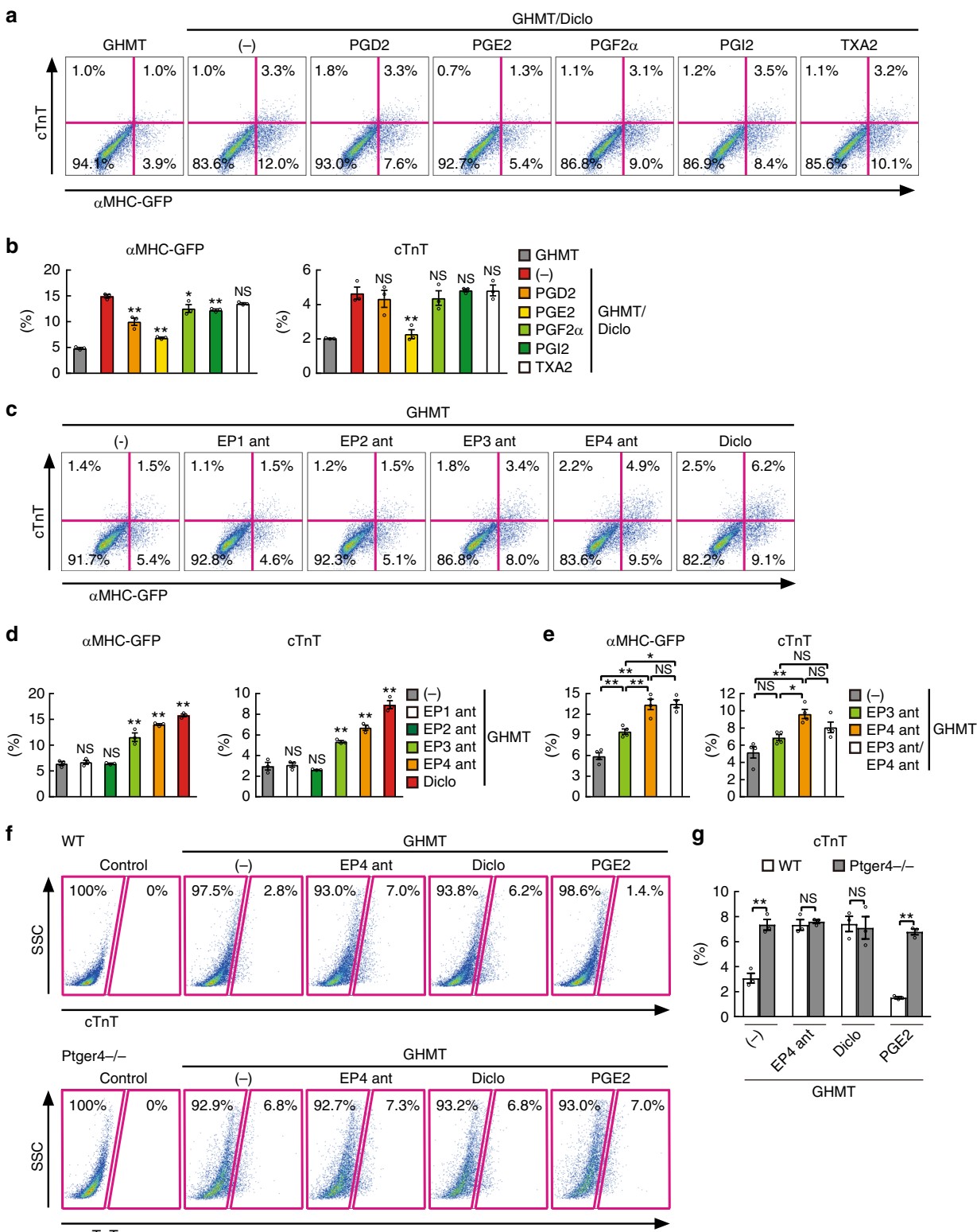

**Fig. 4** Diclofenac promotes cardiac reprogramming via inhibition of the PGE2/EP4 pathway. **a**, **b** FACS analyses for αMHC-GFP and cTnT expression. Postnatal αMHC-GFP TTFs were transduced with GHMT and cultured with the indicated prostanoids for 1 week. Quantitative data are shown in **b**; $n = 3$ biologically independent experiments. **c**–**e** FACS analyses for αMHC-GFP and cTnT expression. GHMT-transduced postnatal TTFs were cultured with the indicated EP antagonists (ant) for 1 week. Quantitative data are shown in **d** and **e**; $n = 4$ biologically independent experiments. See also Supplementary Figure 3. **f**, **g** FACS analysis for cTnT expression in GHMT-treated TTFs from WT or *Ptger4*$^{-/-}$ mice with or without EP4 antagonist, diclofenac, or PGE2. Quantitative data are shown in **g**; $n = 4$ biologically independent experiments. Student's *t*-test was performed for **g**, one-way ANOVA with Dunnett's post hoc test was performed for **b**, **d**, one-way ANOVA with Tukey's post hoc test was performed for **e**; all data are presented as mean ±SEM *$P < 0.05$, **$P < 0.01$ vs. the relevant control. NS, not significant. Diclo, diclofenac

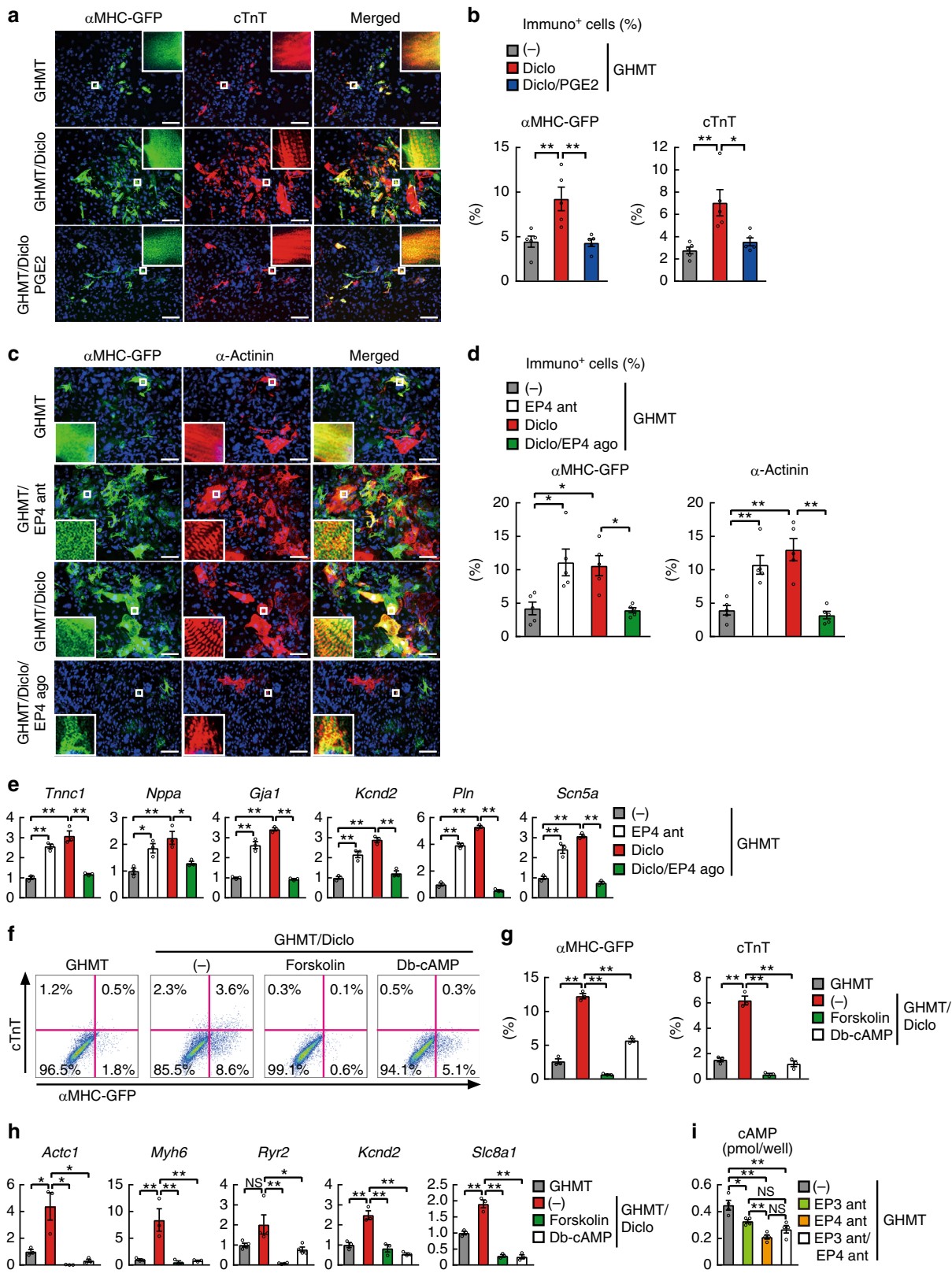

while dibutyryl-cAMP (db-cAMP) is an analog of cyclic AMP that stimulates PKA signaling. We found that these two activators of the cAMP/PKA pathway both counteracted diclofenac-mediated cardiac reprogramming in postnatal and adult TTFs, as shown by FACS and qRT-PCR (Fig. 5f–h, Supplementary

Fig. 4d-f). These results suggest that diclofenac promotes cardiac reprogramming, at least in part, via the inhibition of the cAMP/ PKA pathway. Consistent with this, the intracellular cAMP level was strongly downregulated by EP4 antagonist (Fig. 5i). EP3 couples with both Gs and Gi to activate and inhibit adenylyl

**Fig. 5** Activation of cyclic AMP/protein kinase A pathway inhibits diclofenac-induced cardiac reprogramming. **a**, **b** Immunocytochemistry for αMHC-GFP, cTnT, and DAPI. PGE2 treatment inhibited GHMT/diclofenac-induced cardiac protein expression in postnatal TTFs after 4 weeks. High-magnification views, insets, show the sarcomeric organization. Quantitative data are shown in **b**; n = 5 biologically independent experiments. See also Supplementary Figure 4. **c**, **d** Immunocytochemistry for αMHC-GFP, α-actinin, and DAPI. EP4 antagonist (ant) and diclofenac treatment increased generation of iCMs, while EP4 agonist (ago) inhibited GHMT/diclofenac-induced cardiac reprogramming in TTFs after 4 weeks. High-magnification views, insets, show the sarcomeric organization. Quantitative data are shown in **d**; n = 5 biologically independent experiments. **e** qRT-PCR analyses for cardiac gene expression in postnatal TTFs transduced with GHMT and treated with or without indicated reagents for 1 week; n = 3 biologically independent experiments. **f**, **g** FACS analyses for αMHC-GFP+ and cTnT+ cells. GHMT-transduced TTFs were cultured with diclofenac with or without forskolin or dibutyryl-cAMP (Db-cAMP) for 1 week. Quantitative data are shown in **g**; n = 3 biologically independent experiments. **h** qRT-PCR analyses for cardiac gene expression in postnatal TTFs treated as indicated for 1 week; n = 3 biologically independent experiments. **i** Concentration of intracellular cAMP in GHMT-transduced TTFs treated with the indicated EP antagonists (ant); n = 4 biologically independent experiments. One-way ANOVA with Tukey's post hoc test was performed for **b**, **d**, **e**, **g–i**. All data are presented as mean ±SEM. *P < 0.05, **P < 0.01 vs. the relevant control. NS, not significant. Diclo, diclofenac. Scale bars represent 100 μm

cyclase/cAMP signaling, respectively[21,22]. We found that EP3 antagonist mildly repressed cAMP expression, and addition of EP3 antagonist to EP4 antagonist did not further downregulate the cAMP level, consistent with the cardiac reprogramming efficiency induced by EP3 and EP4 antagonists (Fig. 4c–e, Supplementary Fig. 3b). Thus, EP3 and EP4 antagonists may inhibit the same cAMP pathway to promote cardiac reprogramming.

**Diclofenac suppressed inflammation and fibroblast signatures.** To investigate the mechanisms responsible for diclofenac-mediated cardiac reprogramming, we used microarray analyses to determine the global gene expression profiles of GHMT-transduced TTFs with and without diclofenac, after 1 week. We also analyzed the gene expression in GHMT-TTFs treated with an EP4 antagonist (ONO-AE3-208), as the addition of the EP4 antagonist showed similar effects on cardiac reprogramming as diclofenac did. Differential gene expression analyses revealed that, compared to GHMT alone, 290 genes were upregulated after either diclofenac or EP4 antagonist treatment, while 241 genes were downregulated at least twofold, with significant overlaps between the two groups (Fig. 6a, b, d). Gene ontology (GO) analyses showed that the upregulated genes in both diclofenac and EP4 antagonist treatment were significantly enriched for GO terms associated with cardiac function and development, while the downregulated genes were significantly enriched for GO terms associated with fibroblast signatures (extracellular matrix and collagen fibril organization) and inflammatory responses (positive regulation of response to stimulus and cytokine production) (Fig. 6c, e). Microarray data revealed that a panel of cardiac genes were upregulated, while fibroblast and inflammatory genes, such as *Tgfbi*, *Col1a2*, *Fbln1*, *Mcpt8*, *Il1r1*, and *Tnfsf4*, were downregulated with diclofenac (Fig. 6f). We next investigated whether PGE2 supplementation could globally counteract diclofenac-mediated gene regulation. We found that, of the 218 genes upregulated by diclofenac, 209 were suppressed by PGE2 supplementation. In contrast, 139 of the 161 genes downregulated by diclofenac, were upregulated by PGE2, suggesting that most parts of the transcriptional changes affected by diclofenac were mediated via the suppression of PGE2 synthesis (Fig. 6g).

Previous studies showed that TGFβ and Wnt inhibitors promoted cardiac reprogramming[11,12,23]. We asked whether diclofenac promoted cardiac reprogramming through TGFβ or Wnt pathway inhibition. FACS analyses showed that the induction of αMHC-GFP+/cTnT+ iCMs was significantly stronger upon diclofenac treatment than upon TGFβ (A83-01) or Wnt inhibitor (IWR-1) treatment (Fig. 6h, i). Addition of diclofenac to TGFβ or Wnt inhibitor further increased cardiac reprogramming efficiency (Fig. 6h, i), suggesting that diclofenac

promotes cardiac reprogramming via mechanisms other than inhibition of the TGFβ or Wnt signaling pathways. We next analyzed the time course of cardiac reprogramming required for each molecule to promote the generation of iCMs. Sequential FACS analyses revealed that diclofenac started to increase iCM generation after just 2 days, whereas TGFβ and Wnt inhibitors enhanced iCM generation after 7–10 days, suggesting that diclofenac promoted cardiac reprogramming more rapidly and efficiently than the other molecules (Fig. 6j). We next analyzed the global gene expression profiles after treatment with EP4 antagonist, TGFβ inhibitor, and Wnt inhibitor by microarray. GO term analyses revealed that all three reagents upregulated 51 genes related to cardiac function and muscle structures, whereas EP4 antagonist specifically suppressed 203 genes associated with inflammatory and fibroblast signatures (Supplementary Fig. 5a-d). These results suggest that EP4 antagonist enhances cardiac reprogramming via suppression of inflammation and the fibroblast program in postnatal TTFs.

**Suppression of IL-1β/IL-1R1 improved cardiac reprogramming.** COX-2/PGE2/EP4 activates inflammatory responses, including the IL-1β (*Il1b*), MCP-1 (*Ccl2*), and IL-6 (*Il6*) pathways. We found that *Il1b*, its receptor IL-1R1 (*Il1r1*), and *Ccl2*, but not *Il6*, were strongly suppressed by EP4 antagonist (Fig. 7a). To determine the molecular mechanism underlying PGE2/EP4-mediated cardiac reprogramming, each cytokine (IL-1β, MCP-1, or IL-6) was added to GHMT/EP4 antagonist-treated TTFs. Addition of IL-1β, but not MCP-1 nor IL-6, suppressed cardiac reprogramming, suggesting EP4 antagonist promoted cardiac reprogramming via suppression of IL-1β/IL-1R1 signaling (Fig. 7b, c). Addition of IL-1β upregulated multiple inflammatory and fibroblast genes, such as *Ccr1*, *Cxcl3*, *Il33*, *Col12a1*, *Snai1*, and *Timp1*, and downregulated cardiac gene expression as shown by qRT-PCR (Fig. 7d). Intriguingly, COX-2 selective inhibitor (NS-398) strongly suppressed *Il1b*, while COX-1 selective inhibitor (SC-560) did not, consistent with the specific effect of COX-2 inhibitors on cardiac reprogramming (Figs. 3f–k and 7e).

Next, to determine whether suppression of IL-1β/IL-1R1 signaling could improve cardiac reprogramming, we knocked down IL-1R1 expression with siRNA (si-Ilr1) in GHMT-transduced TTFs (Fig. 7f). FACS analyses revealed that knock-down of *Il1r1* greatly increased cardiac reprogramming in postnatal TTFs, recapitulating the effects of EP4 antagonist (Fig. 7g, h). Moreover, addition of si-Il1r1 to EP4 antagonist did not further increase cardiac reprogramming, suggesting EP4 and IL-1R1 signaling acted via the same pathways (Fig. 7i, j). QRT-PCR revealed that a panel of inflammation- and fibroblast-related genes, including *Cxcl11*, *Cxcl14*, *Il33*, *Col1a1*, *Fn1*, and *Postn*, were significantly downregulated, while cardiac genes, such as *Actc1*, *Nppa*, and *Pln*, were upregulated by si-Il1r1 treatment

 9

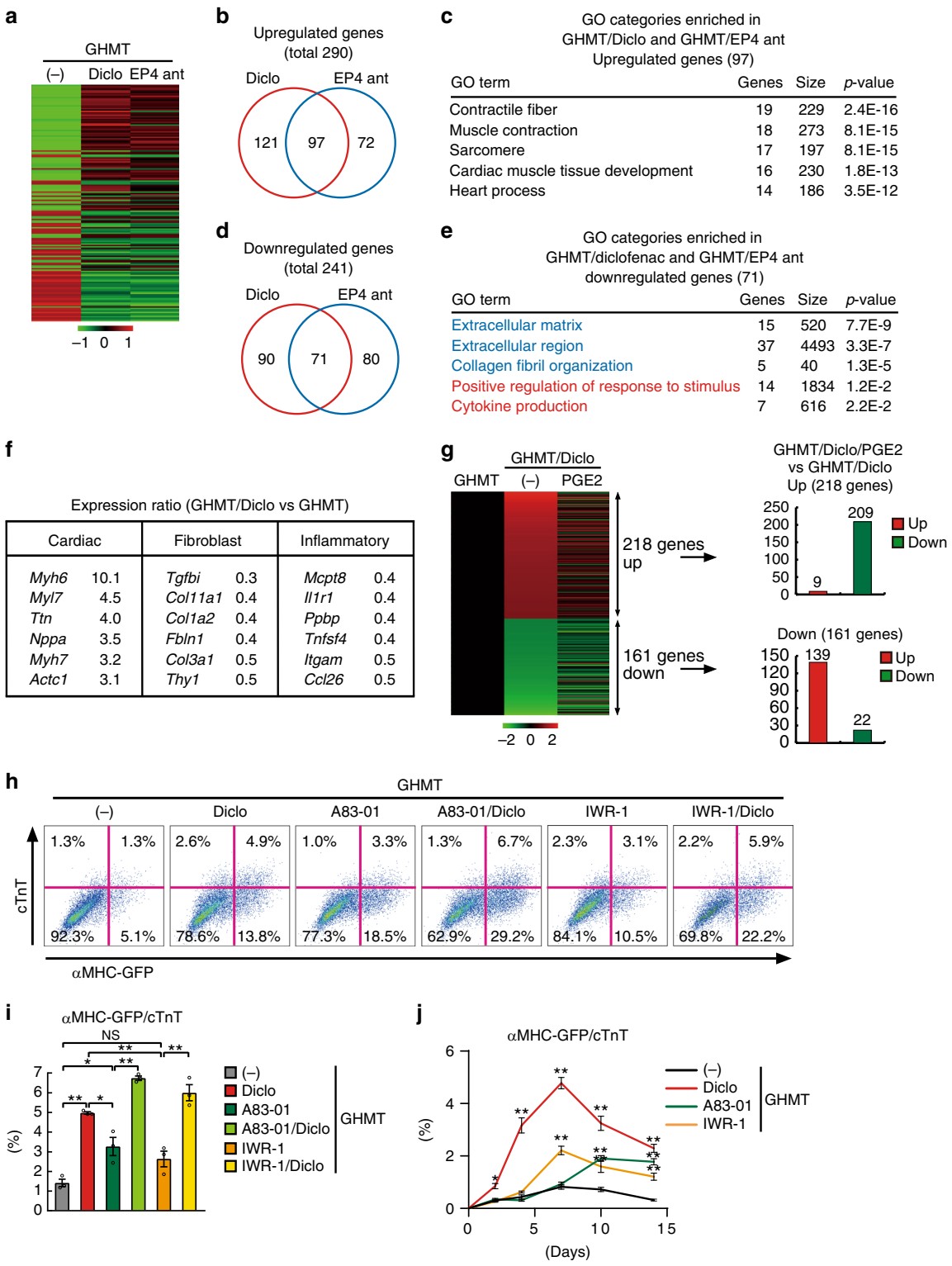

(Fig. 7k). These results suggest that inhibition of the COX-2/PGE2/EP4 pathway promotes cardiac reprogramming via suppression of IL-1β/IL-1R1 signaling and silencing of inflammatory and fibroblast signatures in TTFs.

## Discussion

We have developed a high-throughput, high-content screening system to identify small molecules that promote cardiac reprogramming. We found that a traditional NSAID, diclofenac, greatly promoted cardiac reprogramming in mouse postnatal and adult TTFs, but not in MEFs, by suppressing COX-2/PGE2/EP4 pathways and silencing inflammatory and fibroblast gene expression. To our knowledge, this is the first report identifying the cardiac reprogramming barriers associated with aging.

Previous studies mainly used a candidate approach to identify cardiac reprogramming factors such as transcription factors, miRNAs, growth factors, and small molecules[1,10–13]. Factors were

**Fig. 6** Diclofenac activates cardiac program and suppresses fibroblast and inflammatory gene expression during cardiac reprogramming. **a** Heatmap images of microarray data illustrating differentially expressed genes between either GHMT and GHMT/diclofenac (Diclo) or GHMT and GHMT/EP4 antagonist (ant)-induced TTFs, after 1 week. **b**, **c** Venn diagram showing the genes that are upregulated in postnatal TTFs with GHMT/diclofenac (Diclo, red) or GHMT/EP4 antagonist (ant, blue) by more than twofold compared to GHMT alone. GO analyses of the 97 upregulated genes in both GHMT/diclofenac and GHMT/EP4 antagonist (ant) treatment are shown in **c**. Cardiac-related GO terms are shown. **d**, **e** Venn diagram showing the genes that are downregulated in postnatal TTFs with GHMT/diclofenac (Diclo, red) or GHMT/EP4 antagonist (ant, blue) treatment by more than twofold compared to GHMT alone. GO analyses of the 71 downregulated genes in both GHMT/diclofenac and GHMT/EP4 antagonist (ant) treatment are shown in **e**. Fibroblast and inflammatory-related GO terms are shown. **f** The relative mRNA expression after 1 week of cardiomyocyte, fibroblast, and inflammation genes in GHMT/diclofenac-TTFs compared to GHMT-TTFs by microarray. **g** Heatmap image of microarray data for GHMT-, GHMT/diclofenac-, and GHMT/diclofenac/PGE2-treated TTFs after 1 week of transduction (left panel). Differentially expressed genes between GHMT-TTFs and GHMT/diclofenac-TTFs are shown; 209 out of 218 upregulated genes were suppressed by PGE2 addition (right upper panel), while 139 out of 161 downregulated genes were increased with PGE2 (right lower panel). **h**, **i** FACS analyses for αMHC-GFP$^+$ and cTnT$^+$ cells. GHMT-transduced TTFs were treated as indicated for 1 week. Quantitative data are shown in **i**; $n = 3$ biologically independent experiments. See also Supplementary Figure 5. **j** Time course of FACS analyses for αMHC-GFP$^+$ and cTnT$^+$ cells. Diclofenac induced cardiac reprograming more rapidly than other inhibitors. $n = 4$ biologically independent experiments. One-way ANOVA with Tukey's post hoc test was performed for **i**, two-way ANOVA with Tukey's post hoc test was performed for **j**. All data are presented as mean ±SEM. *$P < 0.05$, **$P < 0.01$ vs. the relevant control. NS, not significant; The scales are –1 to +1 in log$_{10}$ (**a**) and –2 to +2 in log$_{10}$ (**g**). Diclo, diclofenac

selected based on information related to cardiogenesis and cardiac differentiation from PSCs. Although this approach advanced cardiac reprogramming research, improvements in reprogramming were mainly found in MEFs, with limited success in postnatal and adult TTFs[13,16]. In this study, we took an unbiased approach to identify small molecules that promote cardiac reprogramming in postnatal and adult TTFs. We found that diclofenac greatly promoted cardiac reprogramming in mouse postnatal and adult TTFs but not in MEFs. Given that diclofenac and other NSAIDs were not indicated in cardiac differentiation in embryos or from PSCs, our unbiased screening was necessary to identify diclofenac as a cardiac reprogramming-inducing reagent. We anticipate that our screening system will be equally useful in identifying new reagents for cardiac reprogramming in other cell types, such as human aged fibroblasts.

Diclofenac promoted cardiac reprogramming in TTFs via the suppression of COX-2/PGE2/EP4/cAMP/PKA/IL-1β/IL-1R1 signaling, with the inhibition of inflammatory and fibroblast gene expression. COX-2 is a rate-limiting and inducible enzyme of the prostanoid biosynthetic pathway, eliciting inflammation, immune response, and pyrexia[18,19]. We found that not only diclofenac but also other COX-2 specific inhibitors (NS-398 and celecoxib) promoted cardiac reprogramming. In contrast, neither a COX-1 inhibitor (SC-560) nor acetylsalicylic acid substantially improved cardiac reprogramming, suggesting that COX-2 is a main target for diclofenac-induced cardiac reprogramming. Consistently, IL-1β, a barrier for cardiac reprogramming, was suppressed by the COX-2 inhibitor but not by the COX-1 inhibitor. We also found that PGE2 and an EP4 selective agonist both suppressed diclofenac-mediated cardiac reprogramming, whereas the inhibition of EP4 (with a specific antagonist, using siRNA, or in knockout mice) greatly increased cardiac reprogramming efficiency to a similar extent as diclofenac treatment. This suggests that PGE2/EP4 receptor is a major pathway for diclofenac-induced cardiac reprogramming. Consistent with this, microarray analyses revealed that the global transcriptional changes induced by diclofenac significantly overlapped with those induced by EP4 antagonist, which were largely reversed by PGE2 supplementation. A G-protein-coupled receptor, EP4 stimulates adenylyl cyclase via Gs, leading to activation of the cAMP/PKA pathway[18]. The cAMP level was strongly downregulated by EP4 antagonist, and forskolin and db-cAMP, activators of this pathway, suppressed diclofenac-mediated cardiac reprogramming, suggesting that inhibition of the cAMP/PKA pathway is critical for diclofenac/EP4 antagonist-mediated cardiac reprogramming.

We also found that diclofenac induced more efficient and rapid cardiac reprogramming than TGFβ inhibitor or Wnt inhibitor and that addition of diclofenac to TGFβ or Wnt inhibitor-enhanced cardiac reprogramming, suggesting that diclofenac induced cardiac reprogramming through mechanisms other than the inhibition of TGFβ or Wnt signaling. GO term analyses demonstrated that EP4 antagonist, TGFβ inhibitor, and Wnt inhibitor all upregulated cardiac-related genes, whereas EP4 antagonist specifically suppressed inflammation- and fibroblast-related genes in TTFs. Intriguingly, EP4 antagonist suppressed IL-1β and IL-1R1 expression in fibroblasts, and supplementation of IL-1β increased inflammatory and fibroblast genes and inhibited cardiac reprogramming. In contrast, inhibition of IL-1β/IL-1R1 signaling promoted cardiac reprogramming via suppression of inflammatory and fibroblast signatures. These results demonstrated that diclofenac/EP4 antagonist promoted cardiac reprogramming through inhibition of IL-1β/IL-1R1 signaling and suppression of inflammatory and fibroblast programs. Notably, we found that COX-2, EP4, and multiple inflammation- and fibroblast-related genes were more strongly expressed in postnatal and adult TTFs than in MEFs, consistent with the cell type-specific effects of diclofenac on cardiac reprogramming. Thus, active inflammation and subsequent fibrogenesis could be an age-related barrier to cardiac reprogramming. Consistent with this, Zhou et al.[24] recently reported that zinc finger transcription factor 281 (ZNF281) enhanced cardiac reprogramming by modulating cardiac and inflammatory gene expression in adult fibroblasts. Given that the silencing of fibroblast signatures is a prerequisite for reprogramming, anti-inflammation may represent a potential target for other lineage conversions associated with aging[25].

The PGE2/EP4 pathway regulates not only inflammation but also non-inflammatory pathways, such as vascular development and endothelial cell function[26]. Future studies are needed to determine the non-inflammatory mechanisms involved in PGE2/EP4-mediated cardiac reprogramming. We also found that the cardiac reprogramming efficiency in diclofenac-treated adult TTFs was still lower than that in diclofenac-treated postnatal TTFs and that addition of miR-133 promoted cardiac reprogramming in diclofenac-treated adult TTFs to a degree comparable to that in diclofenac-treated postnatal TTFs. Thus, postnatal and adult TTFs are different, and inhibition of inflammatory responses alone may not be sufficient for reprogramming, at least in the context of adult fibroblasts. Identification of epigenetic barriers and other key molecules will provide new insights into the mechanism of cardiac reprogramming and improve reprogramming efficiency in aged

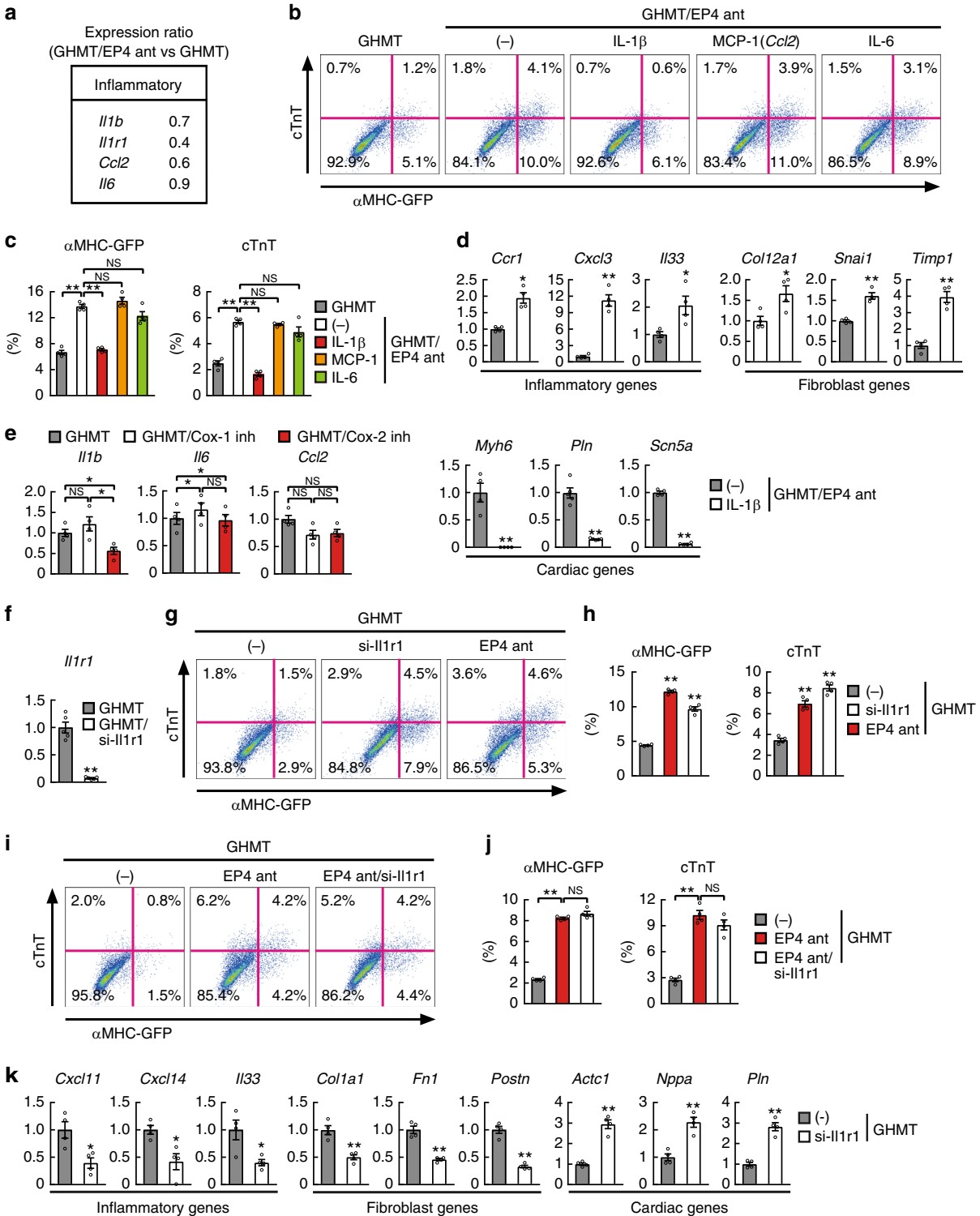

fibroblasts. A recent paper showed that, compared with GMT alone, the delivery of TGFβ and Wnt inhibitors in the presence of GMT improved in vivo cardiac reprogramming and enhanced cardiac repair after MI in mice[23]. It would be interesting to determine whether diclofenac promotes in vivo cardiac reprogramming and improves cardiac function after MI, with TGFβ and Wnt inhibitors for regenerative purposes.

In summary, we developed a high-content screening system to identify small molecules for cardiac reprogramming. We found that diclofenac promoted cardiac reprogramming in postnatal and adult fibroblasts. Given that cardiac diseases affect child and adult patients, our results are of relevance for future applications in disease modeling, drug discovery, and regenerative medicine. Use of the high-content screen will allow for identification of new

**Fig. 7** EP4 antagonist induces cardiac reprogramming via inhibition of IL-1β/IL-1R1 signaling. **a** The relative mRNA expression in GHMT/EP4 antagonist-TTFs compared to GHMT-TTFs by microarray. See also Fig. 6. **b, c** FACS analyses for αMHC-GFP$^+$ and cTnT$^+$ cells. GHMT-transduced TTFs were cultured with EP4 antagonist with or without IL-1β, IL-6, or MCP-1 for 1 week. Quantitative data are shown in **c**; $n = 4$ biologically independent experiments. **d** Relative mRNA expression in GHMT/EP4 antagonist-TTFs with or without IL-1β; $n = 4$ biologically independent experiments. **e** qRT-PCR analyses of inflammatory gene expression in postnatal TTFs treated with COX inhibitors for 1 week; $n = 4$ biologically independent experiments. **f** Relative mRNA expression of *Il1r1* in GHMT-TTFs transfected with scrambled siRNA or si-Il1r1; $n = 5$ biologically independent experiments. **g, h** FACS analyses for αMHC-GFP$^+$ and cTnT$^+$ cells. GHMT-transduced TTFs were cultured with or without si-Il1r1 or EP4 antagonist (ant) for 1 week. Quantitative data are shown in **h**; $n = 4$ biologically independent experiments. **i, j** FACS analyses for αMHC-GFP$^+$ and cTnT$^+$ cells. GHMT-transduced TTFs were cultured with EP4 antagonist (ant) with or without si-Il1r1 for 1 week. Quantitative data are shown in **j**; $n = 4$ biologically independent experiments. **k** Relative mRNA expression of inflammatory genes, fibroblast genes, and cardiac genes in GHMT-TTFs transfected with scrambled siRNA or si-Il1r1; $n = 4$ biologically independent experiments. Student's $t$-test was performed for **d, f, k**, one-way ANOVA with Dunnett's post hoc test was performed for **h**, one-way ANOVA with Tukey's post hoc test was performed for **c, e, j**; all data are presented as mean ±SEM. *$P < 0.05$, **$P < 0.01$ vs. the relevant control. NS, not significant

reagents and advance cardiac reprogramming technology toward clinical applications.

## Methods

**Mouse strains**. The Keio University Ethics Committee for Animal Experiments approved all experiments in this study. Transgenic mice overexpressing *GFP* under the αMHC promoter were generated as described previously[1]. αMHC-Cre/R26-tdTomato mice were generated by crossing αMHC-Cre mice and R26R-tdTomato reporter mice (The Jackson Laboratory)[27]. Generation of EP4-knockout mice (*Ptger4*$^{-/-}$) has been reported previously[28].

**Quantitative and semiquantitative RT-PCR**. Total RNA was isolated from the cells, and a qRT-PCR was performed using SYBR green technology (TOYOBO Life Science) or the Viia7 Real-Time PCR system (Roche or Applied Biosystems) following standard protocols. Details of primers and probes are provided in Supplementary Table 1. Semiquantitative RT-PCR for PGE receptors was performed with following primers: *Ptger1* (Applied Biosystems, Mm00443098_g1), *Ptger2* (Applied Biosystems, Mm00436051_m1), *Ptger3* (Applied Biosystems, Mm01316856_m1), *Ptger4* (Applied Biosystems, Mm00436053_m1). Reactions were conducted in a Veriti 96-well Thermal Cycler (Applied Biosystems). An initial incubation of 10 min at 95 °C was followed by 36 cycles of 1 s at 95 °C and 20 s at 60 °C. PCR products were identified by electrophoresis on a 2% agarose gel and visualized by ethidium bromide staining. The mRNA levels were normalized to those of *Gapdh*.

**Fibroblast isolation**. Fibroblast isolation (of MEFs and postnatal and adult TTFs) was performed as described previously[8]. For MEF isolation, embryos isolated from 12.5-day pregnant mice were washed with phosphate-buffered saline (PBS), followed by careful removal of the head and visceral tissues. The remaining parts of the embryos were washed in fresh PBS, minced using a pair of scissors, transferred to a 0.125% trypsin/ethylenediaminetetraacetic acid (EDTA) solution (Gibco, 25200-072; 3 mL per embryo), and incubated at 37 °C for 20 min. An additional 3 mL of trypsin/EDTA solution was then added, and the mixture was further incubated at 37 °C for 20 min. After trypsinization, an equal amount of medium (6 mL of Dulbecco's modified Eagle medium, DMEM, containing 10% fetal bovine serum, FBS, per embryo) was added and pipetted several times to allow for tissue dissociation. After incubation of the tissue/medium mixture for 5 min 25 °C, the supernatant was transferred to a fresh tube. Cells were collected by centrifugation and resuspended for culturing in DMEM/10% FBS (Thermo Scientific, SV30014.03) at 37 °C in 5% CO$_2$. For TTF isolation, postnatal day 1–3 or 8–week-old adult mouse tails were minced into pieces smaller than 1 mm$^3$ (ref.[12]). The explants were plated on gelatin-coated dishes and cultured for 10–14 days in explant medium (IMDM with L-glutamate and 25 mM HEPES, Gibco, 12440-053, 20% FBS). Fibroblasts that migrated were harvested and filtered with 40 μm cell strainers (BD Biosciences) to avoid contamination with tissue fragments. The MEFs and TTFs were plated at a density of $1 \times 10^4$ cells/cm$^2$ for virus transduction.

**Retroviral infection and cell culture**. To construct pMXs retroviral vectors, we amplified the coding regions of *GFP*, *Gata4*, *Mef2c*, *Tbx5*, and *Hand2* by PCR and subcloned them into respective pMXs vectors for transfection into Plat-E cells using Fugene 6 (Promega, E2691) to generate retroviruses[1]. Fibroblasts were transduced with the retrovirus mixtures as indicated. An siRNA Negative Control (20 μM, Thermo Scientific, 12935-100), siGENOME Mouse *Ptger4* siRNA-SMARTpool (20 μM, Dharmacon, M-048700-01-0005), siGENOME Mouse *Il1r1* siRNA-SMARTpool (20 μM, Dharmacon, M-046644-01-0005), miRIDIAN microRNA Mimic Negative Control #2 (20 μM, Thermo Scientific, CN-002000-01-05), or miRIDIAN microRNA Mouse mmu-miR-133a-Mimic (20 μM, Thermo Scientific, C-310407-07-0005) was transfected simultaneously with retroviruses into cells with Lipofectamine 2000 (Invitrogen, 11668019). The medium was

replaced with FBS medium after 24 h of infection and changed to FFV medium after 2 weeks, as shown previously[12]. FBS medium contained DMEM (high glucose) with L-glutamate and phenol red (Wako, Tokyo, Japan, 044-29765); Medium199 with Earle's Salts, L-glutamate and 22 g/L sodium bicarbonate (Gibco, 11150-059); and 10% Hyclone Characterized FBS (Thermo Scientific, SV30014.03). FFV medium contained StemPro-34 SF medium (Gibco, 10639-011), GlutaMAX (10 μL/mL, Gibco, 35050-061), ascorbic acid (50 μg/mL, Sigma Aldrich, A-4544), recombinant human VEGF165 (5 ng/mL, R&D Systems, 293-VE-050), recombinant human FGF basic 146 aa (10 ng/mL, R&D Systems, 233-FB-025), and recombinant human FGF10 (50 ng/mL, R&D Systems, 345-FG-025). Diclofenac (10 μM, Calbiochem, 287840), SC-560 (10 μM, Cayman, 70340), NS-398 (10 μM, Cayman, 70590), celecoxib (10 μM, Cayman, 10008672), acetylsalicylic acid (10 μM, Sigma Aldrich, A5376), PGD2 (1 μM, Cayman, 12010), PGE2 (1 μM, Cayman, 14010), PGF2α (100 nM, Cayman, 16010), PGI2 (1 μM, Cayman, 18220), TXA2 (1 μM, Cayman, 19600), ONO-8713 (1 μM), TG4-155 (1 μM, Focus Biomolecules, 10-1574), ONO-AE5-599 (1 μM), ONO-AE3-208 (1 μM), ONO-AE1-329 (1 μM), forskolin (10 μM, Enzo Life Science, BML-CN100-0010), Db-cAMP (250 μM, Wako, 023-16381), A83-01 (0.5 μM, Tocris, 2939), IWR-1 (1 μM, Sigma Aldrich, I0161), IL-1β (10 ng/mL, Wako, 094-04681), IL-6 (100 ng/mL, Wako, 093-04433), or MCP-1 (100 ng/mL, Wako, 131-16691) were added simultaneously with retrovirus infection in the indicated experiments. The medium was changed every 3 days.

**High-content screening**. αMHC-Cre/R26-tdTomato TTFs were plated in 96-well plates at a density of 3000 cells per well. Cells were reprogrammed with the GMT retrovirus as described above. The compounds were added simultaneously to the wells to reach to 10 μM of final concentration with a 1:100 dilution of the drug-library. We screened libraries of 8400 toxicologically tested compounds (TMDU chemical library). Among the 96 wells tested, 16 and 80 wells were used for controls (1% DMSO) and chemical compound treatment, respectively. At day 1 of reprogramming, the medium was replaced with FBS medium. After 1 week of reprogramming, cell nuclei were stained with Hoechst 33342, and cardiac reprogramming efficiency was automatically scored as the percentage of αMHC-Cre/R26-tdTomato$^+$ nuclei out of the total nuclei, using an IN Cell Analyzer 6000 (GE Healthcare Life Sciences). Data were analyzed using the IN Cell Analyzer 6000 image-analysis package and normalized as fold change compared to the DMSO control. Hit compounds were defined as those higher than the DMSO control mean + 2 SD.

**FACS analyses**. For αMHC-GFP/cTnT expression, cells were fixed with 4% paraformaldehyde (PFA; MUTO PURE CHEMICALS, Tokyo, Japan, 3311-1) for 15 min and permeabilized with saponin (Sigma Aldrich, 47036-250G-F). They were stained with anti-cTnT (Thermo Scientific, MS-295-P1, 1:100) and anti-GFP (MBL, 598, 1:500) antibodies, followed by secondary antibodies conjugated with Alexa Fluor 488 (Invitrogen, A11008, 1:200) or 647 (Invitrogen, A21240, 1:200)[1]. Cells were then analyzed using a flow cytometer (Beckman Coulter, CytoFLEX S) with CytExpert (Beckman Coulter). As negative controls for FACS gating, cells were stained with isotype control antibodies.

**Immunocytochemistry**. Cells were fixed in 4% PFA for 15 min at room temperature, blocked with 5% normal goat serum blocking solution (Vector Laboratories, S-1000), and incubated with primary antibodies against cTnT (Thermo Scientific, MS-295-P1, 1:100), GFP (MBL, 598, 1:500), or sarcomeric α-actinin (Sigma Aldrich, 111M4845, 1:800). Cells were then incubated with secondary antibodies conjugated with Alexa Fluor 488 (Invitrogen, A11008, 1:200) or 546 (Invitrogen, A11003, 1:200), followed by DAPI (Invitrogen, D1306) counterstaining. The percentage of cells immunopositive for GFP, α-actinin, and cTnT were counted in 10–15 randomly selected fields per well in five independent experiments. A total of 2000–4000 cells were counted. The measurements and calculations were conducted in a blinded manner.

**Ca$^{2+}$ imaging and counting beating cells**. Ca$^{2+}$ imaging was performed following standard protocols. Briefly, cells were labeled with Rhod-3 from a Calcium Imaging Kit (Invitrogen, R10145) for 1 h at room temperature, washed, and incubated for an additional hour, to allow de-esterification of the dye. Rhod-3-labeled cells were analyzed at 37 °C using an All-in-One fluorescence microscope (BZ-9000; Keyence, Tokyo, Japan)[12]. Ca$^{2+}$ oscillations could only be imaged for a short period, owing to the increasing background fluorescence from the medium; thus, the measurements were taken within 30 min after changing to Tyrode's buffer. Ca$^{2+}$ oscillation$^+$ cells were counted in 15 randomly selected fields per well in at five independent experiments.

To count the number of beating cells, we seeded 25,000 fibroblasts per well on 24-well plates, performed cell transductions, cultured the cells, and then monitored cell contractions. For accurate analyses of the cell count, we used the All-in-One fluorescence microscope as described previously[12]. The cells were maintained at 37 °C and 5% CO$_2$, using the controlled chamber within the microscope. We first acquired images of the cells in all the areas in a well with a ×20 phase-contrast lens by moving the motorized stage sequentially. We next moved the field to cover all the areas in a well, and counted the number of spontaneously contracting cells in each field with the ×20 phase-contrast lens in three independent experiments. We identified the individual beating iCMs based on differences in beating frequency, cell membrane boundary, and nuclei identified by the phase-contrast and GFP fluorescence filter. The number of Ca$^{2+}$ transient$^+$ or beating cells were divided based on region of interest areas and shown as the number per cm$^2$. The measurements and calculations were conducted in a blinded manner.

**Microarray analyses**. After 1 week of reprogramming, RNA was extracted from GHMT-, GHMT/diclofenac-, GHMT/EP4 antagonist-, GHMT/diclofenac/PGE2-, GHMT/TGFβ inhibitor-, or GHMT/Wnt inhibitor-induced TTFs using the ReliaPrep RNA Cell Miniprep System (Promega, Z6012). The RNA quality was determined by the RNA Integrity Number (RIN) value with an RNA6000 assay (Agilent). Only specimens with RIN > 7.0 were used in this study. Gene expression levels were determined by microarray (Clariom S Array, Mouse, Affymetrix) according to the manufacturer's instructions. Prior to analysis, all data were normalized by using a Single Space Transformation and Robust Multichip Analysis (SST-RMA) algorithm with Affymetrix Expression Console software version 1.4. Heatmaps of differentially expressed genes (>2-fold) were generated using Gene-Spring GX (Agilent) 14.8 software. Gene ontology (GO) analyses were performed using Database for Annotation, Visualization, and Integrated Discovery (DAVID, URL: http://david.abcc.ncifcrf.gov/) and p values were determined by a modified Fisher's exact test.

**Enzyme-linked immunosorbent assay**. To measure the amount of cAMP produced, postnatal TTFs were plated in 24-well plates at an initial density of $1 \times 10^4$ cells/cm$^2$. EP3 antagonist or EP4 antagonist were added to GHMT-transduced cells and incubated for 30 min. Cell lysates were prepared, and intracellular cAMP was determined according to the manufacturer's instructions using the cyclic AMP ELISA kit (Cayman, 581001).

**Statistical analyses**. Based on published experiments and our experience, data are assumed to be normally distributed. Differences between groups were examined for statistical significance using Student's $t$-tests or one-way or two-way analyses of variance (ANOVA) followed by Dunnett's or Tukey's post hoc test. Differences with $P$-values of <0.05 were regarded as significant.

## Data availability

The data that support the findings of this study are available from the corresponding author upon request. The microarray data reported in this paper has been uploaded to GEO under accession number GSE116227. The source data underlying Fig.1c and Supplementary Figure 3a are provided in the Supplementary Information.

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

## Acknowledgements

We thank Ono Pharmaceutical Co., Ltd. for providing the specific antagonists for EP1 (ONO-8713), EP3 (ONO-AE5-599), and EP4 (ONO-AE3-208), and the EP4-specific agonist (ONO-AE1-329). M.I. was supported by research grants from the Research Center Network for Realization of Regenerative Medicine, Japan Agency for Medical Research and Development (AMED), the Japan Society for the Promotion of Science (JSPS) (17K19678, 15K15313), Senshin Medical Research Foundation, Takeda Science Foundation, and Daiichi-Sankyo Foundation of Life Science. N.M. was supported by research grants from the JSPS(15J07407), Keio University Grant-in-Aid for Encouragement of Young Medical Scientists, the Senshin Medical Research Foundation, the Japan Foundation for Applied Enzymology, and the Basic Research of the Japanese Circulation Society.

## Author contributions

N.M. and M.I. designed the experiments. N.M., K.N., F.T., H.Y., T.S., K.M., M.I., H.K., S.H., H.T., S.K., R.O., Y.S., and S.T. carried out the experiments. N.M., S.S., H.O., and K.F. analyzed the data. N.M. and M.I. wrote the paper.

## Additional information

**Competing interests:** The authors declare no competing interests.

