## [Peer Review File · Nature Communications]

Reviewers' comments:

Reviewer #1 (Remarks to the Author):

This study performed screening of 8,400 small molecules in chemical library using a high-throughput screening system, and put forward Diclofenac as a candidate to be enhancer of cardiac reprogramming in neonatal and adult tail-tip fibroblasts. The authors examined signaling pathway of Diclofenac-mediated cardiac reprogramming using chemical compounds and gene silencing, and claims that inhibition of a COX2-PGE2-EP4 receptor axis is involved in this cardiac reprogramming. This study demonstrated the potential small molecule that enhances cardiac reprogramming in postnatal fibroblasts and the findings might have in future consequences for clinical relevant regeneration therapy. Nevertheless, there are several issues that warrant further consideration:

1. The authors concluded that EP4 is the major receptor involved in Diclofenac-induced cardiac reprogramming. However, the impact of EP4 antagonist seems similar to that of EP3 antagonist. Is there statistical difference between the effect of EP3 antagonist and EP4 antagonist?
2. There was no additive effect of combination of EP3 and EP4 antagonists. These data did not support the concept that EP4 is the major receptor in cardiac reprogramming. These data suggest that EP3 and EP4 receptor share the same downstream signaling pathways. Please clarify the role of these two signaling pathways.
3. Although microarray analysis demonstrated down-regulation of inflammation-related genes in Diclofenac-treated cells, there is no data connecting inhibition of inflammation and cardiac remodeling. Emerging evidence suggest that PGE2-EP4 signaling regulates noninflammatory roles, such as vascular development, bone remodeling, and endothelial cell barrier function (Please see Yokoyama U et al., The prostanoid EP4 receptor and its signaling pathway. *Pharmacol Rev.* 2013 65(3): 1010-52.). The conclusion that inhibition of inflammatory mediator by EP4 signaling inhibition enhances cardiac remodeling was not supported by the data.
4. How do TGF β and Wnt signaling pathways relate to EP4 signaling pathways? It has been reported that EP4 signaling affects not only cAMP-protein kinase A axis (*Pharmacol Rev.* 2013 65(3): 1010-52.). Data connecting TGF β /Wnt and EP4 signaling pathways or discussion are required.
5. Expression profiling of EP receptors would be of interest.
6. Most data were obtained using three samples. Quantitative data should be statistically analyzed in more than four samples in each group.
7. Post-hoc test for ANOVA should be specified in the method section.
8. Some figures are missing y-axis labels.

Reviewer #2 (Remarks to the Author):

By high-throughput screening, Muroaka et al. identified the anti-inflammatory compound diclofenac improved reprogramming of fibroblasts into cardiomyocytes. Authors provide evidence that inhibition of the inflammatory response, PGE2/EP4-associated in particular, has profound effect on reprogramming. However, some questions remained.

Major concern:

1. Administration of diclofenac improves reprogramming efficiency in adult TTFs, however, the % of reprogrammed cells is still much lower than the cells derived from diclo-treated postnatal TTF, despite the fact that adult TTFs showed higher expression levels of COX2 and EP4 (Figure 3e). This suggests that post-natal and adult TTFs are very different and inhibition of inflammatory response may not be sufficient for reprogramming, at least in the context of adult fibroblasts.

2. Microarray analysis suggests that diclo and EP4 antagonist-treated cells show up-regulation of genes associated with cardiac function and reduced fibroblast signatures. What would be the underlying mechanism for such changes? More precisely, how downregulated inflammation is associated with increased cardiomyocyte signature in treated-fibroblasts.

3. Data shown in Figure 6i suggests that the cells-treated with TGF-beta inhibitor alone give rise to higher percentage of alpha-MHC GFP+ cells than the ones treated with diclo. What would be the advantage of using diclo when compared to TGF-beta inhibitor?

4. The authors used siRNA to knock down EP4 expression. However, in order to provide more decisive answer, it is necessary to use EP4 knocked out transgenic mice.

Minor comments:

1. Apart from the COX-1 and COX-2, the rest of genes analysed in figure 3e are not explained.
2. The full name of EP receptors should be given at the first time when it is mentioned.

Reviewer #3 (Remarks to the Author):

Muraoka, et al. present a manuscript regarding a chemical compound which promotes cardiac reprogramming. The study was well-designed and data are convincing. In addition, the authors worked hard to clarify the mechanisms responsible for promoting cardiac reprogramming. The comments are below.

1. The authors may consider changing the title because not only diclofenac sodium but also another Cox2 inhibitor or EP4 antagonist promoted cardiac reprogramming.
2. The authors found 4 compounds through their novel screening system, but focused on just one compound and leave the others alone. The readers would be curious about the other 3 compounds; what kind of molecules are they? or do they have synergic effect on cardiac reprogramming? It would be helpful to disclose more information regarding the other 3 compounds.
3. The authors claimed that diclofenac improved cardiac reprogramming in a dose-dependent manner (P. 5, line 103), but the reprogramming rate was similar in any dosages (Fig.2b).
4. The authors concluded that Cox2 inhibitor, but not Cox1, promoted cardiac reprogramming through inhibition of PGE2/EP4; however, Cox1 inhibitor should come down to block the same PGE2/EP4. They should explain the specific effect of Cox2 inhibitor on cardiac reprogramming.

Response to Reviewers' Comments

We are pleased that the reviewers considered the work presented in our manuscript of significant importance and interest to justify the submission of a revised manuscript. We appreciate the reviewers' constructive comments and have addressed the concerns of the reviewers with new data and/or revisions to the text. In the manuscript, these changes are indicated in red. Point-by-point responses to the reviewers' comments are presented below. We hope that the editors and reviewers find this revised manuscript to be responsive to their comments and thus suitable for publication in *Nature Communications*.

To Reviewer 1

We thank this reviewer for their positive review and constructive comments on our manuscript. According to these comments, we have revised the manuscript as follows.

Comment 1. The authors concluded that EP4 is the major receptor involved in Diclofenac-induced cardiac reprogramming. However, the impact of EP4 antagonist seems similar to that of EP3 antagonist. Is there statistical difference between the effect of EP3 antagonist and EP4 antagonist?

Response: We thank the reviewer for this question. We found a stronger and more significant effect of EP4 antagonist compared with that of EP3 antagonist in promoting cardiac reprogramming (Fig. 4e, Supplementary Figure. 3b). We have revised the manuscript accordingly (page 8, lines 8–12 in the revised Results section).

Comment 2. There was no additive effect of combination of EP3 and EP4 antagonists. These data did not support the concept that EP4 is the major receptor in cardiac reprogramming. These data suggest that EP3 and EP4 receptor shear the same downstream signaling pathways. Please clarify the role of these two signaling pathways.

Response: We thank the reviewer for their thoughtful comment. We found that diclofenac promoted cardiac reprogramming, at least in part, via inhibition of the cAMP/PKA pathway. EP4 activates the cAMP/PKA pathway through Gs, whereas EP3 couples with both Gs and Gi to activate and inhibit cAMP signaling, respectively. We analyzed the intracellular level of

cAMP and found that EP4 antagonist strongly suppressed the level of cAMP, while EP3 antagonist mildly repressed cAMP expression. Moreover, addition of EP3 antagonist to EP4 antagonist did not further downregulate the cAMP level, consistent with the cardiac reprogramming results of these antagonists (Fig. 4e, Fig. 5i, Supplementary Fig. 3b). Thus, EP3 and EP4 may share the same downstream signaling pathways to affect cardiac reprogramming, though EP4 exhibited stronger effects. We have revised the manuscript accordingly [page 9, lines 17–23 (Results) and page 13, lines 7–8 (Discussion) in the revised manuscript].

Comment 3. Although microarray analysis demonstrated down-regulation of inflammation-related genes in Diclofenac-treated cells, there is no data connecting inhibition of inflammation and cardiac remodeling. Emerging evidence suggest that PGE2-EP4 signaling regulates noninflammatory roles, such as vascular development, bone remodeling, and endothelial cell barrier function (Please see Yokoyama U et al., The prostanoid EP4 receptor and its signaling pathway. Pharmacol Rev. 2013 65(3):1010-52.). The conclusion that inhibition of inflammatory mediator by EP4 signaling inhibition enhances cardiac remodeling was not supported by the data.

Response: We thank the reviewer for this comment. We performed additional experiments and found that the expression of IL-1 β and its receptor IL-1R1 was strongly suppressed by EP4 antagonist. Supplementation with IL-1 β , but not the cytokines IL-6 or MCP-1, inhibited EP4 antagonist-mediated cardiac reprogramming via the upregulation of inflammatory and fibroblast signatures. In contrast, inhibition of IL-1R1 by siRNA promoted cardiac reprogramming by suppressing inflammatory and fibroblast gene expression, recapitulating the effects of diclofenac and EP4 antagonist. Given that the silencing of fibroblast signatures is critical for cardiac reprogramming, EP4 antagonist enhanced cardiac reprogramming mainly through the inhibition of IL-1 β /IL-1R1 signaling and suppression of inflammatory and fibroblast programs (Fig. 7). We also discuss the roles of noninflammatory signaling in PGE2/EP4-mediated cardiac reprogramming and cite the paper recommended by the reviewer in the revised manuscript. We have revised the manuscript accordingly [page 11, lines 5–27 (Results) and page 13, line 16–page 14, line 4 (Discussion) in the revised manuscript].

Comment 4. How do TGF β and Wnt signaling pathways relate to EP4 signaling pathways? It has been reported that EP4 signaling affects not only cAMP-protein kinase A axis (Pharmacol Rev. 2013 65(3):1010-52.). Data connecting TGF β /Wnt and EP4 signaling pathways or discussion are required.

Response: We thank the reviewer for this comment. We performed microarray analyses to identify changes in the global gene expression profiles after treatment with EP4 antagonist, TGF β inhibitor, and Wnt inhibitor. GO term analyses revealed that all three reagents upregulated 51 genes related to cardiac function and muscle structures, whereas EP4 antagonist specifically suppressed 203 genes associated with inflammatory and fibroblast signatures (Supplementary Fig. 5a-d). These new results suggest that EP4 antagonist enhances cardiac reprogramming by suppressing inflammatory and fibroblast programs, which is a specific effect of EP4 antagonist. We have revised the manuscript accordingly [page 10, line 24–page 11, line 3 (Results) and page 13, lines 11–16 (Discussion) in the revised manuscript].

Comment 5. Expression profiling of EP receptors would be of interest.

Response: We thank the reviewer for this comment. Semiquantitative RT-PCR revealed that all four EP receptors were expressed in TTFs, among which EP4 was the most abundantly expressed (Supplementary Fig. 3a). We have revised the manuscript accordingly (page 8, lines 4–5 in the revised Results section).

Comment 6. Most data were obtained using three samples. Quantitative data should be statistically analyzed in more than four samples in each group.

Response: We thank the reviewer for their comment. We performed additional experiments with more than four samples in each group for statistical analysis. We have revised the manuscript accordingly (page 24, line 2; page 25, line 23–25; page 26, line 20-21; page 27, line 25–page 28, line 19 in the revised Figure Legends section).

Comment 7. Post-hoc test for ANOVA should be specified in the method section.

Response: We thank the Reviewer for this comment. We used Tukey's post hoc test following ANOVA. We have revised the manuscript accordingly (page 20, line 2 in the revised Methods section).

Comment 8. Some figures are missing y-axis labels.

Response: We thank the reviewer for this comment. We have corrected the relevant figures in the revised manuscript (Fig. 6i).

To Reviewer 2

We are grateful to the reviewer for their positive review of our manuscript and for the helpful comments and suggestions, which we have used as a basis for revising and improving our manuscript as follows.

Major concern:

Comment 1. Administration of diclofenac improves reprogramming efficiency in adult TTFs, however, the % of reprogrammed cells is still much lower than the cells derived from diclo-treated postnatal TTF, despite the fact that adult TTFs showed higher expression levels of COX2 and EP4 (Figure 3e). This suggests that post-natal and adult TTFs are very different and inhibition of inflammatory response may not be sufficient for reprogramming, at least in the context of adult fibroblasts.

Response: We thank the reviewer for this comment. The cardiac reprogramming efficiency in diclofenac-treated adult TTFs was indeed lower than that in diclofenac-treated postnatal TTFs. However, new data showed that addition of miR-133 promoted cardiac reprogramming in diclofenac-treated adult TTFs to a comparable degree as that in diclofenac-treated postnatal TTFs. Thus, as postnatal and adult TTFs are different, inhibition of inflammatory responses alone may not be sufficient for cardiac reprogramming in adult fibroblasts (Fig. 2a, b, Fig. 3c, d, Supplementary Fig. 2a, b). Identification of miRNAs, epigenetic barriers, and other key molecules will improve the reprogramming efficiency in aged fibroblasts. We have revised the manuscript accordingly [page 6, lines 22–25 (Results) and page 14, lines 4–11 (Discussion) in the revised manuscript].

Comment 2. Microarray analysis suggests that diclo and EP4 antagonist-treated cells show up-regulation of genes associated with cardiac function and reduced fibroblast signatures. What would be the underlying mechanism for such changes? More precisely, how downregulated inflammation is associated with increased cardiomyocyte signature in treated-fibroblasts.

Response: We thank the reviewer for this thoughtful comment. We found that the expression of IL-1 β and its receptor IL-1R1 was strongly suppressed by EP4 antagonist. Supplementation with IL-1 β , but not the cytokines IL-6 or MCP-1, inhibited EP4 antagonist-mediated cardiac reprogramming via the upregulation of inflammatory and fibroblast signatures. In contrast, inhibition of IL-1R1 by siRNA promoted cardiac reprogramming by suppressing inflammatory

and fibroblast gene expression, recapitulating the effects of diclofenac and EP4 antagonist. Given that the silencing of fibroblast signatures promotes cardiac reprogramming, diclofenac/EP4 antagonist enhanced cardiac reprogramming mainly through inhibition of IL-1 β /IL-1R1 signaling and suppression of inflammatory and fibroblast programs (Fig. 7). We have revised the manuscript accordingly [page 11, lines 5–27 (Results) and page 13, line 16–page 14, line 4 (Discussion) in the revised manuscript].

Comment 3. Data shown in Figure 6i suggests that the cells-treated with TGF-beta inhibitor alone give rise to higher percentage of alpha-MHC GFP+ cells than the ones treated with diclo. What would be the advantage of using diclo when compared to TGF-beta inhibitor?

Response: We thank the reviewer for this comment. α MHC-GFP is a marker of cardiomyocytes, but bona-fide cardiomyocytes express both α MHC-GFP and cTnT. New data showed that the induction of α MHC-GFP⁺/cTnT⁺ iCMs was significantly stronger with diclofenac treatment than with TGF β inhibitor (A83-01) or Wnt inhibitor (IWR-1), and addition of diclofenac to TGF β or Wnt inhibitor further increased the cardiac induction (Fig. 6h, i). These results suggest that diclofenac is more powerful than other compounds and that diclofenac promotes cardiac reprogramming via mechanisms other than the inhibition of either the TGF β or Wnt signaling pathways, as shown in the new microarray data (Supplementary Fig. 5). We have revised the manuscript accordingly [page 10, line 20–page 11, line 3 (Results) and page 13, lines 11–16 (Discussion) in the revised manuscript].

Comment 4. The authors used siRNA to knock down EP4 expression. However, in order to provide more decisive answer, it is necessary to use EP4 knocked out transgenic mice.

Response: We agree with the reviewer's comment and performed additional experiments using EP4-knockout mice (*Ptger4*^{-/-}). We found that cardiac reprogramming efficiency was greatly increased in *Ptger4*^{-/-} TTFs to a level comparable to that in EP4 antagonist/diclofenac-treated wild-type TTFs. Addition of EP4 antagonist, diclofenac, or PGE2 did not change cardiac induction in *Ptger4*^{-/-} TTFs, suggesting EP4 is critical for diclofenac/PGE2-mediated cardiac reprogramming (Fig. 4f, g). We have revised the manuscript accordingly [page 8, lines 18–24 in the revised Results section].

Minor comments:

Comment 1. Apart from the COX-1 and COX-2, the rest of genes analysed in figure 3e are not explained.

Response: We thank the reviewer for this comment. We have explained the rest of the genes analyzed in Fig. 3e. in the revised manuscript [page 7, lines 11–13 (Results) and page 13, lines 22–24 (Discussion) in the revised manuscript].

Comment 2. The full name of EP receptors should be given at the first time when it is mentioned.

Response: We thank the reviewer for this suggestion. We have provided the full names of all EP receptors the first time they are mentioned (page 8, line 4 in the revised Results section).

To Reviewer 3

We greatly appreciate the reviewer's positive review and their constructive comments on our manuscript. We have revised the manuscript as follows according to the provided comments.

Comment 1. The authors may consider changing the title because not only diclofenac sodium but also another Cox2 inhibitor or EP4 antagonist promoted cardiac reprogramming.

Response: We agree with the reviewer's comment and have changed the title to **“Role of Cyclooxygenase-2/Prostaglandin E2/Prostaglandin E Receptor 4 Signaling in Cardiac Reprogramming”** in the revised manuscript.

Comment 2. The authors found 4 compounds through their novel screening system, but focused on just one compound and leave the others alone. The readers would be curious about the other 3 compounds; what kind of molecules are they? or do they have synergic effect on cardiac reprogramming? It would be helpful to disclose more information regarding the other 3 compounds.

Response: We thank the reviewer for this comment. The functions of the other three compounds remain largely unknown. However, we found that addition of each of the three compounds to diclofenac did not result in additive effects on cardiac reprogramming, suggesting that these three compounds regulate the same signaling pathways as diclofenac (Supplementary Fig. 1a, b). We have revised the manuscript accordingly (page 5, lines 9–12 in the revised Results section).

Comment 3. The authors claimed that diclofenac improved cardiac reprogramming in a

dose-dependent manner (P. 5, line 103), but the reprogramming rate was similar in any dosages (Fig.2b).

Response: We appreciate this comment and examined the effect of diclofenac on cardiac reprogramming at lower dosages (from 1 nM to 100 μ M). These data confirmed that diclofenac improved cardiac reprogramming in a dose-dependent manner and that diclofenac was efficient even at the low concentration (1 μ M). We have revised the manuscript accordingly (page 5, lines 20–23 in the revised Results section).

Comment 4. The authors concluded that Cox2 inhibitor, but not Cox1, promoted cardiac reprogramming through inhibition of PGE2/EP4; however, Cox1 inhibitor should come down to block the same PGE2/EP4. They should explain the specific effect of Cox2 inhibitor on cardiac reprogramming.

Response: We thank the reviewer for their thoughtful comment. We found that *Ptgs2* (COX-2) was strongly expressed in postnatal and adult TTFs compared to levels in MEFs in an age-dependent manner, with expression only barely detectable in postnatal heart samples (Fig. 3e). In contrast, *Ptgs1* (COX-1) expression was 2 to 3-fold higher in postnatal and adult TTFs than in MEFs and heart samples. Moreover, we found that diclofenac/EP4 antagonist enhanced cardiac reprogramming mainly through inhibition of IL-1 β /IL-1R1 signaling and suppression of inflammatory and fibroblast programs (Fig. 7). Intriguingly, COX-2 selective inhibitor (NS-398) strongly suppressed IL-1 β expression, while COX-1 selective inhibitor (SC-560) did not, consistent with the specific effect of COX-2 inhibitors on cardiac reprogramming (Fig. 3f-k). We have revised the manuscript accordingly [page 7, lines 6–11 and page 11, lines 14–16 (Results); page 12, lines 26–27 (Discussion) in the revised manuscript].

We believe that we have appropriately addressed the concerns raised by the reviewers with new data or with revisions to the text. Their insightful comments have enabled us to significantly improve the manuscript. We thank the reviewers for their time, and hope the revised manuscript is now acceptable for publication in *Nature Communications*.

Response to Reviewers' Comments

We are pleased that the reviewers considered the work presented in our manuscript of significant importance and interest to justify the submission of a revised manuscript.

We appreciate the reviewers' constructive comments and have addressed their concerns with new data and revisions to the text. In the manuscript, these changes are indicated in red. Point-by-point responses to the reviewers' comments are presented below. We hope that the editor and reviewer find this revised manuscript to be responsive to their comments and thus suitable for publication in *Nature Communications*.

To Reviewer 2

We greatly appreciate the reviewer's positive review of our manuscript. We have revised the manuscript as follows according to the provided comments.

Comment 1. The authors provide data showing the difference between adult TTFs and post-natal TTFs. As these two populations are different, it is confusing why some experiments were performed using the postnatal TTFs while others were done with the adult TTFs, for instance in Figure 5. It is not clear what the rationale of using two different populations is and what the message this study wants to convey when one population was chosen for a specific experiment.

Response: We thank the reviewer for this question. We found that diclofenac promoted cardiac reprogramming via the PGE2/EP4 receptor pathway in both postnatal and adult TTFs, demonstrating the universal effects and associated signaling pathway of diclofenac on cardiac reprogramming. Although using two different populations of TTFs in all experiments would be ideal, it is highly labor intensive; therefore, we analyzed only one population in some experiments. However, following the reviewer's suggestion, we newly determined the involvement of the cAMP/PKA pathway in PGE2/EP4-mediated cardiac reprogramming in adult TTFs (Supplementary Fig. 4d-f). We have revised the manuscript accordingly (page 9, lines 17–18 in the revised Results section).

Comment 2. The data show that combination of diclo and TGF-b or Wnt inhibitor improves cardiomyocyte reprogramming in comparison with TGF-b or Wnt inhibitor alone. However, it

remains not clear if diclo is indeed a more powerful agent for cardiomyocyte reprogramming. The % of α -MHC-GFP⁺/cTnT⁺ iCMs in the condition of diclo, TGF- β or Wnt inhibitor alone was not analyzed in the Figure 6i. It will be more convincing to show the time frame that is required for each molecule to promote generation of iCMs.

Response: We thank the reviewer for this thoughtful comment. Based on this suggestion, we analyzed the percentage of α -MHC-GFP⁺/cTnT⁺ iCMs upon treatment with diclofenac, TGF β inhibitor, or Wnt inhibitor alone (Fig. 6i). We also analyzed the time frame required for each molecule to promote the generation of iCMs. We found that diclofenac started to increase iCM generation after just 2 days, whereas TGF β and Wnt inhibitors enhanced iCM generation after 7–10 days, suggesting that diclofenac promoted cardiac reprogramming more rapidly and efficiently than TGF β or Wnt inhibitors (Fig. 6j). We have revised the manuscript accordingly (page 10, line 23–page 11, line 5 in the revised Results section).

We believe that we have appropriately addressed the concerns raised by the reviewers with new data and revisions to the text. We thank the reviewers for their time and hope the revised manuscript is now acceptable for publication in *Nature Communications*.

REVIEWERS' COMMENTS:

Reviewer #2 (Remarks to the Author):

The authors have successfully addressed my questions with additional new results.

REVIEWERS' COMMENTS:

Reviewer #2 (Remarks to the Author):

The authors have successfully addressed my questions with additional new results.

Response to Reviewers' Comments

We are pleased that the reviewers find our revised manuscript acceptable. We once again thank the reviewers for their time and constructive comments.